# Diverse Counterfactual Explanations for Anomaly Detection in Time Series

## Abstract

Data-driven algorithms for detecting anomalies in times series data are ubiquitous, but generally unable to provide helpful explanations for the predictions they make. In this work we propose a post-hoc explainability method that is applicable to any differentiable anomaly detection algorithm for time series. Our method provides explanations in the form of a set of diverse counterfactual examples, i.e., multiple perturbed versions of the original time series that are similar to the latter but not considered anomalous by the detection algorithm. Those examples are informative on the important features of the time series and the magnitude of changes that can be made to render it non-anomalous for the explained algorithm. We call our method *counterfactual ensemble explanation*, and test it on two deep-learning-based anomaly detection models. We apply the latter to univariate and multivariate real-world data sets and assess the quality of our explanations under several explainability criteria such as *Validity, Plausibility, Closeness* and *Diversity*. We show that our algorithm can produce valuable explanations; moreover, we propose a novel visualization of our explanations that can convey a richer interpretation of a detection algorithm's internal mechanism than existing post-hoc explainability methods. Additionally, we design a sparse variant of our method to improve the interpretability of our explanation for high-dimensional time series anomalies. In this setting, our explanation is localized on only a few dimensions and can therefore be communicated more efficiently to the model's user.

## 1 Introduction

Anomaly detection in time series is a common data analysis task that can be defined as identifying outliers, i.e., observations that do not belong to a reference distribution. For instance, anomaly detection is leveraged to localize a defect in computing systems, disclose a fraud in financial transactions, or diagnose a disease from health records Blázquez-García et al. (2021). Detected outliers often call for further investigation, therefore, the recipient of a detection algorithm outputs generally needs to be able to interpret the algorithm's predictions. Consequently, providing explanations for models that detect anomalies has practical relevance, all the more in the setting of multivariate time series data, where model interpretation is an even more challenging task. This is however a still understudied problem, in particular for machine learning models.

In general, an anomaly detection model classifies each timestamp of a time series as anomalous or not. Several state-of-the-art models involve complex deep learning (DL) classifiers, such as LSTMs Malhotra et al. (2015), RNNs Audibert et al. (2020) or TCNs Bai et al. (2018); Carmona et al. (2021), whose internal mechanisms are opaque. This lack of transparency can prevent these models from being deployed in consequential contexts Brown et al. (2018); Bhatt et al. (2020). Prior work has proposed to include interpretable blocks in machine learning models for anomaly detection (e.g., attention mechanism in RNNs Brown et al. (2018)) or design model-specific explainability methods (e.g., feature-importance scores for Isolation Forests Carletti et al. (2021)). Our work is orthogonal to these methods: we propose a post-hoc and model-agnostic explainability method that can be applied to any existing differentiable anomaly detection model.

The majority of existing post-hoc explainability methods for time series models aims at estimating feature-saliency scores Crabbe & van der Schaar (2021); Pan et al. (2020). The latter ranks the features of the

input data in terms of their relative contribution to the model's prediction. Although these techniques have provided valuable insights in image classification tasks (Fong et al., 2019), it is often a weak form of explanation for anomalies in time series. In fact, they essentially indicate that the time series values at the anomalous time stamps are salient, therefore providing redundant information compared to the anomaly detection model (see for instance Figure 1b, where the salient features are highlighted in green). In practice, a user of an anomaly detection model might be interested in (a) knowing what can be changed in the input data to avoid encountering the anomaly again in the future (preferentially with minimal cost), and (b) understand the model's sensitivity to a particular anomaly. Our proposed method provides explanations satisfying these two requirements.

Counterfactual explanation denotes a type of explainability method that provides insight on the sensitivity of a model's predictions to a change in the input data. They have notably been proposed for interpreting time series classifiers Ates et al. (2021); Delaney et al. (2021); Karlsson et al. (2020). A counterfactual example (or for short, a counterfactual) is an instance-based explanation in the form of a perturbed input on which the model's prediction value is different from the model output on the original data. It thus indicates what modifications of the input must be made to obtain a different prediction. It is generally defined as an instance $X'$ minimizing a cost function such as Wachter et al. (2018):

$$L(X, X', y', \lambda) = \lambda(f(X') - y')^2 + d(X, X'),$$

where $X$ and $f$ are respectively the original input and the prediction model that need to be explained, $y'$ is a desired output value (e.g. a different predicted label in classification contexts), $d(.,.)$ is a distance on the input space and $\lambda$ is a trade-off parameter. In their basic definition, they are closely related to adversarial examples Verma et al. (2020), however, their properties and their utility are distinct. Adversarial examples are often weakly constrained and used as hard instances to train more robust models, whereas counterfactuals are designed as plausible examples for interpreting an existing model's predictions.

In the context of anomalies detected by a time series model, counterfactual methods aim at generating modified time series (or sub-sequences) that do not contain anomalous observations according to the detection model. With the additional constraint that counterfactuals are somehow similar to the original time series, these time series instances therefore correspond to the closest normal or expected behaviour according to the explained model. For example, when the time series is a temporal record of a patient's blood glucose level with abnormally high values, a counterfactual example can be an alternative record with levels in a non-critical interval. Hence, counterfactual explanations can reveal the boundaries of the normal time series distribution according to the prediction model.

However, a single counterfactual is generally only a partial explanation, satisfying a particular trade-off between predefined criteria Russell (2019). One extension of counterfactual explanation consists in providing an *ensemble* (or set) of *diverse* counterfactual instances Russell (2019); Mothilal et al. (2020); Dandl et al. (2020). Nonetheless, this extension has not been previously considered in the context of time series anomaly detection models. Besides, more broadly, there is no existing strategy to effectively communicate these more complex counterfactual explanations to the model's user. In this work, we propose an approach for generating counterfactual ensemble explanations for anomaly detection models in time series, as well as a visualization method of these explanations.

More precisely, we make the following contributions:

- We introduce a model-agnostic and post-hoc method that explains the predictions of any differentiable anomaly detection model for time series. For any given input and prediction value, our explanation, called *counterfactual ensemble explanation*, is a set of counterfactual examples satisfying different trade-offs between pre-defined criteria. In practice, these examples can be used individually as actionable explanations, or analysed together to investigate the model's sensitivity to perturbations of the input.

- We design a *sparse* variant of our method for high-dimensional time series anomalies, which have been much less studied and generally harder to interpret. In this context, we constraint our counterfactual explanation to make changes only on a few dimensions of the input time series, so that it can be communicated more efficiently to the explanation's recipient.

- We propose an interpretable visualization of our counterfactual ensemble explanation. Our representation shows the range of possible perturbations gaining insight on the model's local decision boundary and sensitivity. Thus, our visualization can increase the actionability of the counterfactual explanation, when the time series features are mutable.

- We investigate the value of our method on two deep-learning anomaly detection models, applied to univariate and multivariate real-world time series data sets. We quantify the quality of our ensemble explanations using metrics previously proposed in other data domains, namely *Validity, Plausibility, Closeness* and *Diversity*. We note that ensemble explanations have never been considered in the context of time series anomalies, therefore there is not yet an equivalent competitive method. However, we also design a *naive* counterfactual ensemble method that we numerically compare to in our experiments.

Figure 1 illustrates our proposed method, and its novelty in contrast to existing explainability methods for time series models. In this univariate example with a spike outlier, a feature-saliency explanation method essentially highlights the time series features near the anomaly (Figure 1b). Besides, a (single) counterfactual explanation proposes a whole new subsequence where the largest feature changes are localized at the anomaly (Figure 1c). In comparison, our ensemble explanation (Figure 1d) is (a) *sparse*, in the sense that it is localized on a few time series features (the anomaly) (b) *optimal* in that it minimally modifies these features and (c) *rich* by diversifying the possible perturbations (see Figure 1e, showing a few examples from our ensemble).

After succinctly reviewing existing work in explainability for time series models and counterfactual explanations in Section 2, we describe the general set-up in Section 3. In Section 4, we present our approach. Then in Section 5, we demonstrate the effectiveness of our method on DL-based models and benchmark anomaly detection data sets. Finally, we discuss our results and propose possible future developments in Section 6.

## 2 Related work

Explainability methods for users of machine learning models have developed along two paradigms: building models with interpretable blocks or designing model-agnostic methods that can be applied to any model already deployed. For time series data, RETAIN Choi et al. (2016) incorporates an attention-mechanism in an RNN-based model while Dynamic Masks Crabbe & van der Schaar (2021) is a model-agnostic algorithm that produces sparse feature-importance masks on time series using dynamic perturbation operators. In fact, many methods for time series adapt algorithms designed for tabular or image data: for instance, TimeSHAP Bento et al. (2021) extends SHAP, a feature-attribution method that approximates the local behaviour of a model with a linear model using a subset of features. Another interesting line of work interprets CNNs for time series models using Shapelet Learning Ma et al. (2020). Shapelets are subsequences that are learnt from a dataset to build interpretable time series decompositions.

Nonetheless, previously cited work for time series are feature-saliency estimation methods. Although they are notably helpful to localize the important parts of time series (in terms of their contribution to the model's prediction), they can only weakly explain anomaly detection models. Moreover, *instance- or example-based* explanations can be more easily interpreted by a non-expert person Wachter et al. (2018). These methods explain a prediction on a single instance by comparing it to another real or generated example, e.g., the most typical examplar of the observed phenomenon (a *prototype* Hautamaki et al. (2008)) or a contrastive examplar related to a distinct behaviour (a *counterfactual* Ates et al. (2021); Delaney et al. (2021); Karlsson et al. (2020)). For time series classifiers, counterfactuals can be generated by swapping the values of the most discriminative dimensions with those from another training instance Ates et al. (2021). In a causal inference setting, Chernozhukov et al. (2021) construct counterfactual time series as linear combinations of control groups. Unfortunately, these approaches can yield implausible subsequences, that do not belong to the data manifold Carletti et al. (2021), e.g., by breaking correlations between the dimensions of multivariate time series. The Native Guide algorithm Delaney et al. (2021) does not suffer from the previous issue but uses a perturbation mechanism on the Nearest Unlike Neighbor in the training set using the model's internal feature vector. Lastly, for a k-NN and a Random Shapelet Forest classifiers, Karlsson et al. (2020) design a tweaking mechanism to produce counterfactual time series.

However, these methods necessitate knowledge of the model's internal mechanism and/or access to its training dataset, which can be expensive. Additionally, these counterfactual explanations suffer from the so-called *Rashomon effect* Molnar (2019), i.e., the fact that several equally-good perturbed examples might exist and be informative for the model's user. In this case, one might benefit from knowing multiple ones, before choosing the most helpful example in a specific context Mothilal et al. (2020). For linear classifiers of tabular data, a set of diverse counterfactuals can be obtained by sequentially adding constraints along the optimization iterations of the perturbation algorithm Russell (2019), whereas the Multi-Objective Counterfactuals algorithm Dandl et al. (2020) records multiple perturbed examples generated along the iterations of a genetic algorithm. These counterfactual sets therefore contain different trade-offs between conflicting criteria. While in the previous methods, diversity is not explicitly enforced, the DiCE algorithm Mothilal et al. (2020) includes a penalization on counterfactuals' similarity based on Determinantal Point Processes. In a similar fashion, for image classifiers, DiVE Rodriguez et al. (2021) perturbs the latent features in a Variational Auto Encoder and penalises pairwise similarity between perturbations, while Karimi et al. (2020) propose a general framework for generating counterfactual examples with diversity constraints in heterogeneous data. Our paper differs from these works since it considers the problem of generating diverse counterfactual explanations for the time series domain. In particular, we leverage specific time series perturbation mechanisms in order to obtain plausible examples.

To the best of our knowledge, we propose the first method that provides diverse counterfactual explanations for time series. As previously noted Crabbe & van der Schaar (2021), this data domain requires specific treatment of temporal dependencies, therefore existing methods for tabular data cannot be directly applied. Besides, having a diverse set of counterfactual explanations can be particularly helpful for time series where the actionable or mutable features are not known in advance. We introduce our method in the context of anomaly detection, however we believe that our approach could be adapted to other tasks on time series data. Moreover, previous works proposing diverse counterfactual explanations have not discussed the additional challenge of communicating efficiently a set of examples compared to a single one. The visual representation we propose can be related to the "What-If Tool" Wexler et al. (2020), an interactive visual tool designed for general ML model elicitation. Before exposing our method, we describe the general set-up in the next section.

## 3 General set-up

In this work, we assume that anomalies in a time series are unpredictable and out-of-distribution subsequences. Hence, an anomaly is a significant deviation from a given reference behaviour. In the remainder, we will not make a distinction between anomaly, outlier and anomalous/abnormal/atypical observation. Not-anomalous data points will be considered as belonging to the data distribution, and denoted as the reference/normal/typical/expected behaviour. We will also refer to the latter as the *context*.

For the description of the general set-up, we introduce the following notations: for an integer $k \in \mathbb{N}$, $[k]$ denotes the set $\{i;\ 1 \le i \le k\}$ and for $x \in \mathbb{R}$, let $x_+ = \max(0, x)$. For a vector $v \in \mathbb{R}^n$, we denote $v_i$ its $i$-th coordinate and for $X \in \mathbb{R}^{m \times n}$ a matrix or multivariate time series, $X_i$ denotes respectively the $i$-th row or the $i$-th observation.

### 3.1 Anomaly detection model

We assume that we are given an anomaly detection model which we can use to predict anomalies on a time series of any given length. We consider a general setting where time series are multivariate and the model processes all dimensions (or *channels*) jointly. More precisely, we denote $X \in \mathbb{R}^{T \times D}$ a time series with $T$ time stamps and $D$ dimensions. The prediction function of the model, denoted by $f$, is used to classify each timestamp $t \in [1, T]$ of $X$ as "anomalous" (i.e., label 1) or "not-anomalous" (i.e., label 0). In fact, the prediction $f(X) \in \mathbb{R}^T$ is a vector of anomaly scores for each timestamp (e.g., probability scores of being anomalous) which transforms into a vector of 0-1 labels using the model's classification rule (e.g. a threshold on these scores). Note that the dimension of the vector $f(X)$ might be smaller than $T$ if the model needs a warm-up interval.

In practice, these models often detect anomalous time stamps by subdividing time series into smaller time windows and classifying the latter (therefore each timestamp or a subset of them in these sub-windows). In other works, to output a prediction on a single timestamp, the "receptive field" of a model is generally a fixed-size (typically small) window. Let's denote $W \in \mathbb{R}^{L \times D}$ a window of size $L$ and consider the following general set-up: the window $W = [W_C, W_S]$ is subdivided by the model into two parts, with $W_C \in \mathbb{R}^{(L-S) \times D}$ a *context* part (that can be empty if the context is implicit once the model is trained) and $W_S \in \mathbb{R}^{S \times D}$ a *suspect* part, for which the model makes a prediction. More precisely, $f(W) \in \mathbb{R}^S$ is the anomaly score of the window $W_S$ and, without loss of generality, we suppose that $f(W) \in [0,1]^S$. We also denote $\theta \in [0,1]$ the anomaly detection rule, i.e., a label 1 is given to $W_S$ if for some $i \in [S]$, $f(W)_i > \theta$.

Examples of anomaly detection models with the previously described mechanism are NCAD Carmona et al. (2021), where the context window has typically thousands of time stamps and the suspect window has 1 to 5 time stamps, and USAD Audibert et al. (2020), where $W = W_S$ and $L = 5$ or 10. In the latter case, the context is implicit and the whole training set is considered as normal data and thus the context of anomalies detected in a test time series.

### 3.2 Counterfactual explanation

In most cases, a single anomaly is a short subsequence, and can therefore be contained in one or few contiguous subwindows $W_S$. For ease of exposition, we suppose that an anomaly is contained in one suspect window. An example is shown in Figure 1a where a suspect window $W_S$ (highlighted in red) contains an anomaly. A counterfactual example for model $f$ detecting an anomaly in $W_S$ (i.e., for some $i \in [S]$, $f(W)_i > \theta$), is an alternative window $\widetilde{W} = [W_C, \widetilde{W}_S]$ such that all predicted labels are 0 (i.e., for any $i \in [S]$, $f(\widetilde{W})_i < \theta$). Since the context of the anomaly is also key to its detection by the model, and if $W$ does not contain a context window $W_C$, we choose to add in the counterfactual example $\widetilde{W}$ a fixed size window $W_C$, that immediately precedes $W$ in the time series. Note that we implicitly suppose that anomalies are not too close to each other so that the additional context window does not contain any anomaly. With a slight abuse of notations, we still denote $\widetilde{W}$ the obtained counterfactual example.

### 3.3 Properties of counterfactual explanations

There are four largely consensual properties that convey value and utility to counterfactual explanations in the context of model elicitation Verma et al. (2020):

1. *Validity* or *Correctness*: achieving a desired model output, e.g., changing the predicted class label in classification; this is the key goal of a contrastive explanation.

2. *Parsimony* or *Closeness*: minimally and sparsely changing the original input; this is motivated by practical feasibility of the counterfactual if the input features are actionable, and by readibility of the information communicated to the model's user.

3. *Plausibility*: counterfactual explanations need to contain realistic examples of normal subsequences.

4. *Computational efficiency:* being computable within a reasonable amount of time and with acceptable computing resources.

In the context of an anomaly detected in a time series, property (1) is equivalent to flipping the anomaly detection model's prediction label from 1 to 0 (i.e., achieving a anomaly prediction score below the classifier threshold). Property (2) can be enforced by restricting the perturbation of the input on a small window containing the anomaly (i.e., the suspect window $W_S$) and on few dimensions of the time series (if the anomalous features are only located on some channels). Property (3) requires that the counterfactual belongs to the normal data distribution. If the latter is not known or estimated, this criterion can be complicated to evaluate, but some prior knowledge such as the time series' regularity, seasonality, or bounds can be leveraged. Property (4) potentially depends on the specific setting, in particular the cost of using the model's prediction function or its gradient, and the size of the dataset. However, in our context, we assume that accessing the

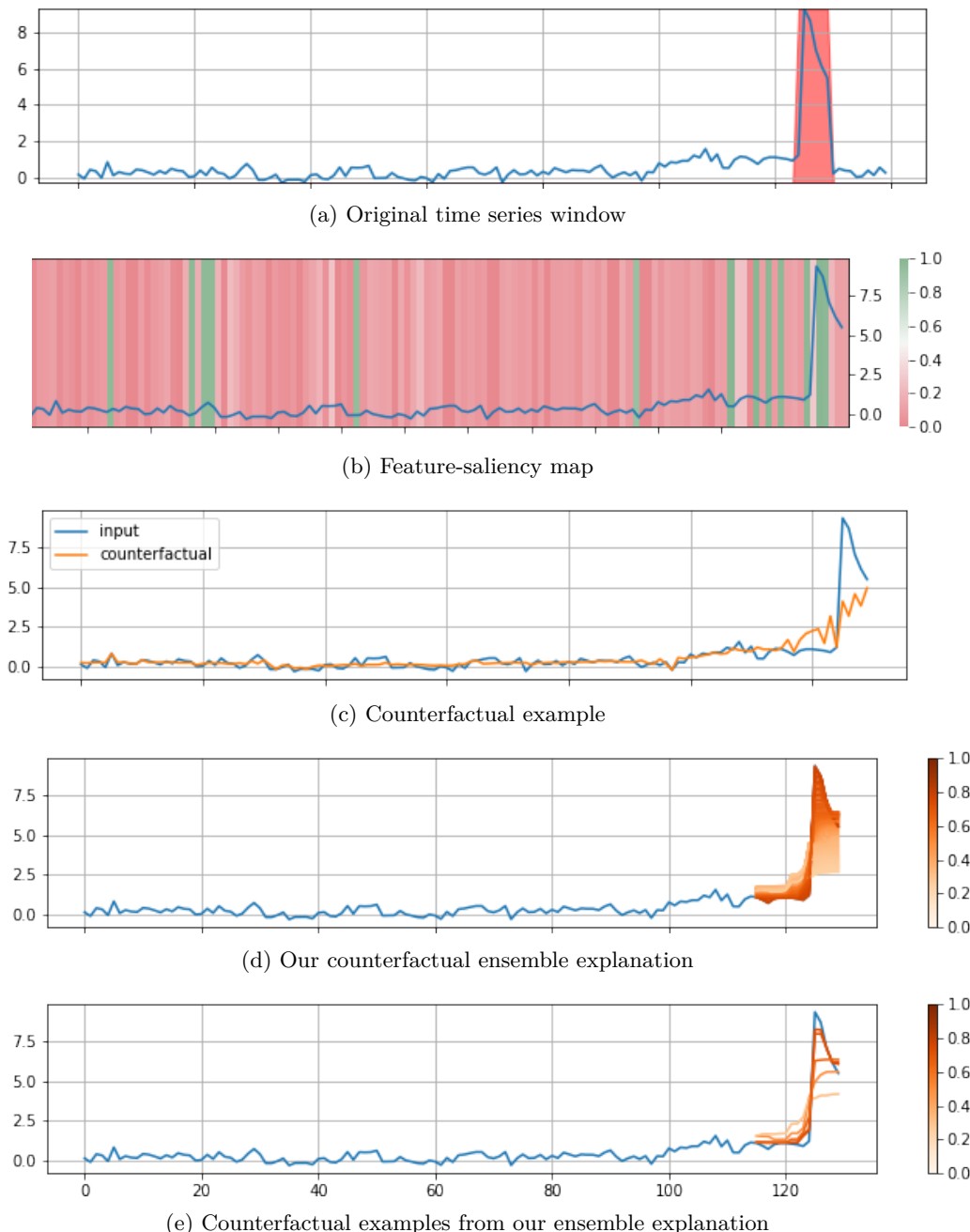

(a) Original time series window

(b) Feature-saliency map

(c) Counterfactual example

(d) Our counterfactual ensemble explanation

(e) Counterfactual examples from our ensemble explanation

Figure 1: Comparison between existing explainability methods for time series and ours, in the context of anomaly detection. The original input (1a) is a univariate time series window containing an anomalous subsequence (a spike outlier, highlighted in red) and the anomaly detection model is NCAD (see Section 5.1.2). The subsequent panels represent the explanations from a feature-importance method (Dynamic Masks Crabbe & van der Schaar (2021)) (1b), an instance-based method (counterfactual example) (1c) and our method (1d). In (1b), the important timestamps have saliency scores closed to one (green color code). In (1d), all the examples from our counterfactual ensemble, which only span the anomalous sub-window, are plotted; the orange color map indicates their anomaly scores (between 0 and 1) given by the explained model. In (1e), we additionally plot five counterfactual examples from this ensemble.

training set of the detection model is particularly expensive, since the latter often decomposes the time series into small windows, leading to a large number of actual training inputs for long time series.

Unfortunately, those properties are often conflicting (e.g., parsimony and plausibility in the context of a spike outlier), therefore a single counterfactual example can only achieve a particular trade-off between them. In the next paragraph, we motivate the use of counterfactual *ensembles* (or sets) as more comprehensive explanations.

### 3.4 Diversity as an additional property

When the data features are actionable , the counterfactual example informs on the localization and magnitude of change that can be applied to the original time series to obtain non-anomalous data. However, the best or feasible trade-off between the pre-defined criteria might depend on the particular anomaly or user's range of action. In absence of this prior knowledge, previous work Mothilal et al. (2020); Russell (2019) added diversity, or range of perturbation, as one informative criterion. In particular, a set of counterfactual examples can increase the likelihood of finding a helpful explanation Rodriguez et al. (2021).

In this sense, an ensemble of counterfactual explanations for an opaque model is more insightful if the user can discriminate between feasible and non-feasible counterfactual examples when given a set of them. However, this qualitative statement is difficult to quantify in practice since there for most data sets, there is no ground-truth for the notion of actionability of counterfactual explanations.

Moreover, we also argue that this additional complexity in the explanation should be adequately communicated to the explanation's recipient, e.g., with a suitable visualization. Intuitively, the latter should be informative on the different possibilities of features changes and the particular trade-off achieved by a counterfactual example. In Section 5.4, we propose a representation for time series, where all counterfactual examples can be visualized together with their anomaly score under the explained model. One example is shown in Figure (1d) and several case studies are represented in Figure 2.

## 4 Methodology

In this section, we present our method for generating counterfactual ensemble explanations. Our approach for differentiable anomaly detection models is described in § 4.1 and can be delineated into two variants, whose respective uses depend on prior knowledge of the data distribution. The first one, called *Interpretable Counterfactual Ensembles (ICEs)* (§ 4.1.1), can be applied without any domain knowledge input. The second one, called *Dynamically Perturbed Ensembles (DPEs)* (§ 4.1.2), leverages dynamic perturbation operators (Crabbe & van der Schaar, 2021), which induce a modification of a time series according to a pre-defined mechanism. Next, we design *sparse* variants of this approach, where the perturbations are restricted to a few dimensions of the input (high-dimensional) time series (§ 4.2). Finally, we describe an alternative method for generating our counterfactual ensemble when the model's gradient information is not available (§ 4.3).

### 4.1 Gradient-based counterfactual ensemble explanations

Most counterfactual algorithms (e.g., Native Guide Delaney et al. (2021), Growing Spheres Laugel et al. (2018), DiCE Mothilal et al. (2020)) rely on adequately perturbing the input $W$ and optimise the perturbation to enforce some properties of the perturbed example. In our method, using the notations of Section 3, we first define an objective function over a single counterfactual example $\widetilde{W} = [W_C, \widetilde{W_S}]$, then use a gradient-descent algorithm starting at the original time series to minimize it. The ensemble of examples is built along the optimization path by collecting adequate perturbations. We define two variants of our method: one, called *Interpretable Counterfactual Ensemble* (ICE), that is a completely unspecified, and another one, *Dynamically Perturbed Ensemble* (DPE), where one can input some domain knowledge and specify a dynamic perturbation mechanism Crabbe & van der Schaar (2021).

### 4.1.1 Interpretable Counterfactual Ensemble (ICE)

In this variant, the objective function on a counterfactual example is defined as follows:

$$\mathcal{L}_{ICE}(\widetilde{W}) = \mathcal{L}_{pred}(\widetilde{W}) + \mathcal{L}_c(\widetilde{W}) + \mathcal{L}_s(\widetilde{W}), \tag{1}$$

where the first term accounts for the Validity property via a hinge loss on the prediction score on $\widetilde{W}$, i.e.,

$$\mathcal{L}_{pred}(\widetilde{W}) = (f(\widetilde{W}) - c)_+,$$

with $c \in [0, 1]$ is a margin parameter. The second term in equation 1 enforces the Closeness constraint via a penalty similar to the elastic net Zou & Hastie (2005), here using the Frobenius and the $L_1$ matrix distances:

$$\mathcal{L}_c(\widetilde{W}) = \frac{\lambda_1}{S\sqrt{D}}\|\widetilde{W} - W\|_1 + \frac{\lambda_2}{SD}\|\widetilde{W} - W\|_F,$$

where $\lambda_1, \lambda_2 > 0$ are regularization parameters. Finally, the third term of equation 1 enforces Plausibility through temporal smoothness (see for instance Crabbe & van der Schaar (2021)):

$$\mathcal{L}_s(\widetilde{W}) = \frac{\lambda_T}{(S-1)D}\sum_{i=1}^{D}\sum_{t=1}^{S-1}|[\widetilde{W}_S]_{(t+1)i} - [\widetilde{W}_S]_{ti}|,$$

with $\lambda_T > 0$. The assumption behind this constraint is that normal time series are not too rough and smoother than abnormal windows, therefore realistic perturbations should also be quite smooth.

### 4.1.2 Dynamically Perturbed Ensemble (DPE)

In this variant, one can specify the perturbation mechanism to obtain the counterfactual ensemble using a dynamic perturbation operator Crabbe & van der Schaar (2021) and a map that spatially and temporally modulates this perturbation. This notably allows to specify the lengthscale of change in the perturbation operator. More precisely, a map is a matrix $M \in [0, 1]^{S \times D}$ that accounts for the amount of change applied to a timestamp and a dimension in the suspect window $W_S$. A value close to 1 in $M$ indicates a big change while a value close to 0 indicates a small change. Here, the dynamic perturbation operator is a Gaussian blur which takes as input a time series window $W$, a timestamp $t \in [L - S, L]$, a dimension $i \in [D]$ and a weight $m \in [0, 1]$, and is defined as:

$$\pi_G(W, t, i, m) = \frac{\sum_{t'=1}^{L} W_{t'i}\exp(-(t-t')^2/2(\sigma_{max}(1-m))^2)}{\sum_{t'=1}^{L}\exp(-(t-t')^2/2(\sigma_{max}(1-m))^2)},$$

with $\sigma_{max} \geq 0$, a hyperparameter tuning the blur's temporal bandwidth. We note that the bigger this parameter is, the larger is the smoothing effect of the perturbation. The latter is called *dynamic* in the sense that it modifies a timestamp using its neighbouring times. We also refer to Crabbe & van der Schaar (2021) for more examples of dynamic perturbation operators.

Finally, for a given map $M$, a perturbed suspect window is given by $[\widetilde{W}_S(M)]_{ti} = \pi(W, L-S+t, i, 1-M_{ti})$, $t \in [S], i \in [D]$. The objective function is then written in terms of the perturbation map as:

$$\mathcal{L}_{DPE}(M) = \mathcal{L}_{pred}(\widetilde{W}(M)) + \frac{\lambda_1}{S\sqrt{D}}\|M\|_1 + \frac{\lambda_2}{SD}\|W - \widetilde{W}(M)\|_F$$

$$+ \frac{\lambda_T}{(S-1)D}\sum_{i=1}^{D}\sum_{t=1}^{S-1}|M_{(t+1)i} - M_{ti}|, \tag{2}$$

where the first term is the hinge loss, and the second and fourth terms account for the sparsity and smoothness constraints, in this case applied on $M$ rather than $\widetilde{W}$ as in equation 1.

---

**Algorithm 1** Gradient-based counterfactual ensemble explanation algorithm.

---

**Input:** The anomalous time series window $W$, the anomaly detection model $f$, the anomaly threshold $\theta$, the learning rate $\eta$, the number of iterations $T$, the number of counterfactual examples $N$.

$\tilde{W}^0 = W$

$I = \{\}$

**for** $t = 1, \ldots, T$ **do**

 Do one step of Stochastic Gradient Descent $\tilde{W}^t = \tilde{W}^{t-1} - \eta \nabla \mathcal{L}(\tilde{W}^{t-1})$

 **if** $\forall i \in [S], \, f(\tilde{W}^t)_i < \theta$ **then**

  Add $\tilde{W}^t$ to I

 **end if**

**end for**

$J = |I|/N.$

Subsample every $J$-th elements of $I$.

**Output:** The set of $N$ counterfactual examples $I$.

---

### 4.1.3 Optimization and complexity.

Our algorithm for differentiable models has the following steps (see also our pseudo-code in Algorithm 1). We first initialize the counterfactual $\tilde{W}$ at the original anomalous window $W$. Then, we minimize the objective function equation 1 or equation 2 using $T$ iterations a Stochastic Gradient Descent (SGD) algorithm. At each iteration $t = 1, \ldots, T$, we evaluate the anomaly detection model at the current value $\widetilde{W}^t$ and if $\forall i \in [S], f(\widetilde{W}^t)_i < \theta$, we add $\widetilde{W}^t$ to a set $I$. After $T$ iterations, we subsample $N$ counterfactuals from the set $I$ to obtain a diverse counterfactual ensemble. In practice, by choosing $T$ around 1000, $N$ around 20-30, and an adequate learning rate, the size of the set $I$ will be much larger than $N$ and for simplicity, we regularly subsample $I$, ordered by the iteration rank of the examples. Complexity-wise, our method therefore requires to query the anomaly detection model and its gradient at each iteration of the SGD algorithm.

We note that in our method, we do not select only the global optimum of our objective functions, but we collect a set of examples along the optimisation path, as long as these examples are non-anomalous. Our heuristic is that by initializing at the original time series, we hope to collect counterfactual examples that are close to the original time series, for a large range of hyperparameters values. Moreover, since defining an *optimal* counterfactual given the Closeness and Plausibility criteria for each anomaly is not easy to specify, the different examples found along the optimization path achieve different trade-offs between the terms in the objective function. In fact, these examples can be seen as solutions of optimization problems with different sets of weights (hyperparameters) in this objective. Note that similar strategies to ours have been previously used for generating ensemble of counterfactuals in distinct data domains, e.g., in Dandl et al. (2020); Russell (2019); Ley et al. (2022).

**Other potential candidates for enforcing diversity.** We now discuss other candidates from literature for enforcing diversity in counterfactual explanations. There are three notable alternative strategies: a) define an objective function over a set of counterfactual examples and include a proximity penalty between the examples, as in Mothilal et al. (2020); Ley et al. (2022); b) select the optima of our objective function for $N$ sets of hyperparameters (e.g., chosen over a grid); and c) select the optima of our objective function for $N$ random initialization points of our algorithm. For strategy a), solving such an objective is much more cumbersome for a large number of features and counterfactual examples. In fact, Ley et al. (2022) note that using a Determinantal Point Process penalty like in Mothilal et al. (2020) requires expensive computations of matrix determinants. Besides, using instead a penalty based on pairwise distances like in Bhatt et al. (2021) may be particularly challenging for time series where non-standard distances must be computed. As for strategy b), solving the optimization problem for $T$ sets of hyperparameters would be much less computationally efficient, and in practice, $T$ would need to be much larger than what we use in our method to obtain $N$ valid counterfactuals since most of the hyperparameter configurations would fail. Finally, additionally to being less computationally efficient, we found that strategy c) is not enough to enforce diversity and often leads to redundant solutions in our experiments. This empirical observation has

been previously noted by Ley et al. (2022) in a different data context and may be due to the fact that a fixed set of hyperparameters induces a "strong" minimum of the objective function.

## 4.2 Sparse counterfactual explanations for high-dimensional time series

A high-dimensional time series would result in a similarly high dimensional explanations. On the other hand, prior work argues that humans prefer simpler explanations Miller (2019). Therefore, one may obtain a simpler explanation by restricting the counterfactual ensemble explanation to span as few dimensions as possible. In this case, the explanation can be more easily visualized and the counterfactual is more actionable, since it then requires to change a minimal number of channels. Besides, anomalies often tend to be concentrated on few dimensions, for instance, when a small subsample of monitoring metrics take abnormal values in a servers network Su et al. (2019b)). Therefore, explanations for these anomalies should also reflect their low-dimensional property. For these reasons, we design a sparse version of our gradient-based method that constraints the counterfactual ensemble explanation to be *spatially* sparse (i.e., sparse or parsimonious in the perturbed dimensions).

### 4.2.1 Sparse ICE

In the sparse version of ICE, we restrict the number of perturbed dimensions by introducing a vector $w \in [0,1]^D$ and a matrix $Z \in \mathbb{R}^{S \times D}$, and defining $\widetilde{W}_S(w, Z) = (w \otimes \mathbf{1}) \odot Z + ((1-w) \otimes \mathbf{1}) \odot W_S$. The role of $w$ is to select the dimensions in $W_S$ that are perturbed with $Z$. We then consider an objective function in terms of $(Z, w)$:

$$\begin{aligned}
\mathcal{L}_{ICE,SP}(w, Z) = &(f(\widetilde{W}(w, Z)) - c)_+ \\
&+ \frac{\lambda_1}{\sqrt{D}}\|w\|_1 + \frac{\lambda_2}{SD}\|W - \widetilde{W}(w, Z)\|_F \\
&+ \frac{\lambda_T}{(S-1)D} \sum_{i=1}^{D} \sum_{u=1}^{S-1} |Z_{(u+1)i} - Z_{ui}|.
\end{aligned} \tag{3}$$

Contrary to equation 1, where the sparsity penalization is applied globally (i.e., both temporally and spatially), the previous objective enforces spatial sparsity through the $L_1$-penalisation on $w$. Another way to see that is to re-interpret objective equation 1 as objective equation 3 with $w = (1, 1, \ldots, 1)$, $Z = \widetilde{W}_S$ and replace the $L_1$-penalisation on $w$ by $\frac{\lambda_1}{S\sqrt{D}}\|Z - W_S\|_1$.

### 4.2.2 Sparse DPE

We apply the same idea to the DPE variant by enforcing the perturbation maps to be spatially sparse. More precisely, we define $M(w, t) = t \otimes w$ with $w \in [0,1]^D$ and $t \in [0,1]^T$ and a loss function in terms of $(w, t)$:

$$\begin{aligned}
\mathcal{L}_{DPE,SP}(w, t) = &(f(\widetilde{W}(w, t)) - c)_+ \\
&+ \frac{\lambda_1}{\sqrt{D}}\|w\|_1 + \frac{\lambda_2}{SD}\|W - \widetilde{W}(w, t)\|_F \\
&+ \frac{\lambda_T}{S-1} \sum_{u=1}^{S-1} |t_{u+1} - t_u|.
\end{aligned} \tag{4}$$

Here the smoothness constraint is applied on $t$ to guarantee that $M$ is also smooth in the temporal dimension.

## 4.3 Gradient-free approach: Forecasting Set

If the anomaly detection model is non-differentiable, we propose an alternative algorithm that generates a counterfactual ensemble explanation using an appropriate sampling mechanism. The pseudo-code for this approach is given in Algorithm 2. We describe the steps in detail here. Machine learning models for time series data sometimes rely on sampling in the context of probabilistic forecasting. Here, we will train an

---

**Algorithm 2** Gradient-free counterfactual ensemble explanation algorithm.

---

**Input:** The anomalous time series window with its context subwindow $W = [W_C, W_S]$, the anomaly detection model $f$, the anomaly threshold $\theta$, the training data $\mathcal{D}$, a probabilistic forecasting model $g$, $W = [W_C, W_S]$, the number of draws $T$.

    Train the model $g$ to predict on $\mathcal{D}$

    Obtain the predictive distribution $g(W_C)$

    $I_{FS} = \{\}$

    **for** $t = 1, \ldots, T$ **do**

        Sample from $g(W_C)$: $W_F^{(t)} \sim g(W_C)$

        **if** $\forall i \in [S], f([W_C, W_F^{(t)}])_i < \theta$ **then**

            Add $\tilde{W}^t = [W_C, W_F^{(t)}]$ to $I_{FS}$

        **end if**

    **end for**

**Output:** The set of counterfactual examples $I_{FS} = \{\}$.

---

auxiliary probabilistic forecasting method and use it as a generative model of counterfactual subsequences. More precisely, given an input window $W_C \in \mathbb{R}^{L-S \times D}$, our auxiliary model $g$ outputs a distribution over a forecast horizon of $S$ time stamps, $g(W_C)$, from which one can sample forecasting paths. We therefore sample $T$ windows $W_F^{(t)} \sim g(W_C)$, $t \in [T]$, then select the ones that are not anomalous according to the anomaly detection model, i.e., our counterfactual ensemble is given by:

$$I_{FS} = \{W_F^{(t)}; \; t \in [T] \text{ st } \forall i \in [S], f([W_C, W_F^{(t)}])_i < \theta\}.$$

Note that one could also subsample the set $I_{FS}$ to obtain a fixed number $N$ of examples. Intuitively, since the probabilistic forecasting model is trained to learn the data distribution, it generates realistic forecast samples. However, the sampling model is oblivious to the original input $W_S$ and therefore the forecasting samples are not restricted to be minimally distant from it. Therefore, in this approach, the Closeness and Sparsity properties are not explicitly accounted for. Nonetheless, one could refine this method by selecting the samples which are closer to the original instance. In our experiments, we study the general behaviour of this method without implementing this minor change. In Section 5, we will construct and evaluate this approach with a Feed Forward Neural Network (FFNN) for univariate data and a DeepVAR model Salinas et al. (2019) for multivariate data from the GluonTS package (Alexandrov et al., 2020) [1].

## 5 Experiments

In this section, we test and compare the performances of our method on two differentiable models, and the relative advantages of its five variants (i.e., ICE, DPE, FS, Sparse ICE, Sparse DPE) in multiple contexts. For this analysis, we have considered two DL anomaly detection models, NCAD Carmona et al. (2021) and USAD Audibert et al. (2020), and four benchmark time series datasets. We report in Section 5.4 a qualitative evaluation of our counterfactual ensemble explanations and their visualization, and in Section 5.5, a quantitative analysis under the previously defined criteria. Note that this study does not include a comparison to existing baselines, since counterfactual ensemble explanations have not been previously considered for time series data. Although some algorithms such as DiCE Mothilal et al. (2020) exist in the context of tabular data, we do not use them in our context since perturbation methods are adapted to each data domain Crabbe & van der Schaar (2021). Nonetheless, for the sake of comparison, we also include a naive baseline, which mechanism is described in Section 5.1. Section 5.2 and Section 5.3 provide additional details on the explainability metrics and the hyperparameters selection procedure.

---

[1] https://ts.gluon.ai/stable/ (accessed on September 11th 2022)

### 5.1 Experimental set-up

#### 5.1.1 Datasets

To evaluate our explainability method, we test it on four data sets that are used to benchmark anomaly detection algorithms on time series, see for example Carmona et al. (2021); Su et al. (2019a); Audibert et al. (2020):

- **KPI:** [2] this data set contains 29 univariate time series. It was released in the AIOPS data competition and consists of Key Performance Indicator curves from different internet companies in 1 minute interval.

- **YAHOO:** [3] this data set was published by Yahoo labs and consists of 367 real and synthetic univariate time series.

- **Server Machine Dataset (SMD):** [4] this dataset contains 28 time series with 38 dimensions, collected from a machine in large internet companies Su et al. (2019a).

- **Soil Moisture Active Passive satellite (SMAP):** [5] this NASA data set published by Hundman et al. (2018) contains 55 times series with 25 dimensions.

The main properties of these data sets are summarized in Table 1. These datasets are suitable for evaluating our explainability method since it contains synthetic and real time series anomalies, in diverse time series domains: Key Performance Indicators, server machines, satellite data, etc. We use these datasets since these are commonly used by SOTA anomaly detection methods Carmona et al. (2021); Su et al. (2019a); Audibert et al. (2020). We note that for these data sets, ground-truth labels of anomalies are available. However, this data does not contain additional context or information on the anomalies, consequently there is no ground-truth explanation, *a fortiori* counterfactual example. This is however a common setup in explainability, and, when user studies are not feasible, one needs to resort to proxies for performing a quantitative evaluation Verma et al. (2020). In Section 5.2, we will define our explainability metrics, which have been previously proposed in multiple data domains (see for instance Mothilal et al. (2020) and Verma et al. (2020)).

More precisely, we use the test sets of each dataset, which correspond to the last 50% time stamps of each time series Carmona et al. (2021). When needed, the training and validation sets contain respectively the first 30% and subsequent 20% time stamps. We note that all these datasets have ground-truth anomaly labels on the test set, and in our evaluation, we only compute counterfactual ensemble explanations for the ground-truth anomalies detected by each model (i.e., the *True Positives*).

In practice, our method could be applied on all the detected anomalies, i.e., on both the *True Positives* (TPs) and the *False Positives* (FPs) (i.e., the observations with anomalous predicted labels that are not ground-truth anomalies). However, we consider a practical case where the user is able to analyse only the true anomalies (i.e., the TPs ) and wants to know what changes would render this input non-anomalous. However, we also performed a complementary analysis to test our method on FPs (see Appendix B.2). These experiments indicate that the performance of our method on FPs is better in terms of our explainability metrics than on TPs. One explanation for this empirical observation is that a small perturbation of the original FP anomalies is often enough to find good counterfactual explanations using our method. Since explaining TPs is more challenging than FPs, we focus on TPs in the main text and report the FP experiments in Appendix B.2.

#### 5.1.2 Anomaly detection models

In our experimental evaluation, we have selected two differentiable SOTA models with distinct temporal neural networks mechanisms. The first one, Neural Contextual Anomaly Detection (NCAD) Carmona et al.

---

[2]https://github.com/NetManAIOps/KPI-Anomaly-Detection (accessed on September 11th 2022)
[3]https://webscope.sandbox.yahoo.com/catalog.php?datatype=s&did=70 (accessed on September 11th 2022)
[4]https://github.com/NetManAIOps/OmniAnomaly (accessed on September 11th 2022)
[5]https://github.com/khundman/telemanom (accessed on September 11th 2022)

(2021), uses a temporal convolutional network and subdivides time series into windows that include a context part. The second one, UnSupervised Anomaly Detection (USAD) Audibert et al. (2020), is based on a LSTM Auto-Encoder and predicts anomalies on suspect windows without explicit context windows. Neither of these models are interpretable-by-design, but both have SOTA performances on the benchmark anomaly detection datasets and reasonable training times (around 90 min). Before evaluating our explainability method, we train these models using the procedure described in their respective papers. More details on these models and their detection performance on the benchmark datasets are reported in Appendix A.

### 5.1.3 Naive counterfactual ensemble explanation

As previously noted, there is no existing method for generating an ensemble of counterfactual examples for time series. We therefore propose a simple interpolation baseline that does not require any training nor optimization procedure. The main idea is similar to the Forecasting Set approach, but here the sampling mechanism is "naive". For each tested window $W$ containing an anomaly in $W_S$, we draw a sample by interpolating the anomalous window $W_S$ and a constant window with a random weight. The constant window repeats the observation from the timestamp immediately before the anomaly, i.e., $[W_C]_{L-S}$. Thus, for $i \in [N]$, a sample $\widetilde{W}_S^{naive,i}$ is defined as:

$$\widetilde{W}_S^{naive,i} = w_i W_S + (1 - w_i) X_{-1}, \tag{5}$$

where $w_i \overset{i.i.d.}{\sim} U[0,1]$ and $X_{-1} = [[W_C]_{L-S}, \ldots, [W_C]_{L-S}] \in \mathbb{R}^{S \times D}$. As in Section 4.3, we also select the samples that are not anomalous under the model, i.e., the naive counterfactual ensemble is finally:

$$I_N = \{\widetilde{W}^{naive,i} = [W_C, \widetilde{W}_S^{naive,i}]; \ i \in [N] \text{ st } \forall t \in [S], f(\widetilde{W}^{naive,i})_t < \theta\}$$

## 5.2 Explainability metrics

To evaluate the utility of our method, we compute the following metrics as proxies of the criteria defined in Section 3:

- **Failure rate:** This metric accounts for the Validity or algorithm Correctness criteria. For the gradient-based methods (DPE, ICE and their sparse variants), it is defined as the percentage of times our method fails to output an ensemble of $N$ counterfactual examples. For the Forecasting Set approach and naive sampling baseline, the failure rate corresponds to the rejection rate of the sampling scheme.

- **Distance:** The Closeness criterion is measured in terms of the Dynamic Time Warping (DTW) distances between each example of the counterfactual ensemble and the original anomalous window. The DTW distance is generally more adapted to time series data than the Euclidean distance.

- **Implausibility:** since the Plausibility property is not easy to evaluate without expert knowledge of the particular data domain, we decompose it into the three following proxy metrics that cover different notions of deviation from an estimated normal behaviour:

  - DTW distance to a reference time series, here, the median sample from the Forecasting Set approach **(Implausibility 1)**;
  - Temporal Smoothness **(Implausibility 2)**, defined as

  $$\sum_{i=1}^{D} \sum_{t=1}^{S-1} |[\widetilde{W}_S]_{(t+1)i} - [\widetilde{W}_S]_{ti}|.$$

  - Negative log-likelihood under the probabilistic forecasting distribution $g$, if available **(Implausibility 3)**.

  We compute the latter metrics for each example of the counterfactual ensemble explanation.

- **Diversity:** the range of values spanned in a counterfactual ensemble is evaluated by the variance of the counterfactual examples at each timestamp.

- **Sparsity correctness:** for multivariate time series, if additional information on the anomalous dimensions in the ground-truth anomalies is available, we compute the precision and recall scores of the sparse variants of DPE and ICE in identifying the dimensions to perturb.

### 5.3 Hyperparameters selection

The hyperparameters of our counterfactual explanation method with the gradient-based approaches are selected by testing all configurations of $\lambda_1 = \lambda_2, \lambda_T$ in the set $\{0.001, 0.01, 0.1, 1.0\}$, $\sigma_{max}$ in $\{3, 5, 10\}$ and the learning rate of the SGD algorithm in $\{0.01, 0.1, 1.0, 10.0, 1000.0, 10000.0\}$. As an explainability method can be finely tuned on a particular problem and dataset, the configurations could be evaluated on all the anomalies in the test set. However, for computational time efficiency reasons, we run this evaluation on 100 randomly chosen anomalies, then evaluate the final performance of the chosen configuration on the entire test set. An exception holds for the the SMAP dataset, which contains less than 100 anomalies detected by the models, therefore we run the configurations' evaluation on the whole test set. For each dataset and detection model, we select the set of hyperparameters having the minimal Implausibility 2, given that the failure rate is kept under a pre-defined level, (see Figures 7, 8, 9 and 10 and tables in Appendix D). We note that here focusing on the Implausibility 2 criterion is an arbitrary choice, and one could use instead any other explainability metric. Moreover, we run the SGD algorithm for 1000 iterations and select a maximum of $N = 100$ counterfactual examples along the optimization path. The hyperparameters of the probabilistic models in the Forecasting Set approach are reported in Table 8 in Appendix D. Finally, in order to provide a ready-to-use method, we also suggest a default set of hyperparameters in Table 13 in Appendix D. For all datasets, models and approaches, we use suspect windows of $S = 10$ time stamps and margin parameter $c = 0$.

### 5.4 Qualitative analysis

Similarly to image classification settings Zeiler & Fergus (2013), visualizations in the time series domain can be human-friendly tools to communicate model explanations, in particular in univariate or low-dimensional settings. In our time series anomaly detection context, we propose to visualize our counterfactual ensemble explanation together with the original time series for which a prediction was made, possibly with an added context window (see Section 3) and on a restricted number of channels. Since the anomaly prediction score given by the explained model is a scalar, we can leverage a color scale to indicate the score of each counterfactual example in the ensemble.

On Figure 2, we present a visualization of our method on two anomalies from the KPI dataset, detected by NCAD and USAD. On each panel, we plot a sub-window of the original time series containing anomalous features in the last 10 time stamps, as well as each counterfactual example given by a variant of our method applied to one of the detection model. Each counterfactual only differs at the anomalous features and the color scheme indicates its anomaly score under the explained model. We argue that this representation allows to deem the range of time series values and prediction scores spanned by the different counterfactuals in our ensemble explanation, and therefore effectively informs on the model's sensitivity and local decision boundary.

We can then visually compare the different variants of our method and the explanations for two detection models. We observe that the counterfactual ensemble explanation from DPE (in red color scale), ICE (in green), and FS (in purple) are quite dissimilar, although they all globally lessen the amplitude of the spike outliers' features. In fact, on the one hand, DPE produces counterfactual ensembles that are less diverse than the other approaches, and relatively close to the original input. This is coherent with the fact that the perturbations are constrained by the dynamic mechanism. On the other hand, ICE's counterfactual sets cover a much larger range of values and therefore allows to visualize more clearly how the anomaly score evolves for different magnitudes of the spikes. This explanation may thus be more informative here since it spans a larger range of time series values. In contrast, the counterfactual ensembles generated by FS do not have the aforementioned interpretation but seem to visually correspond to the expected behaviour given the

shape of the context windows. The previous preliminary observations seem consistent for the two models and confirmed on several other anomalies (see for instance the additional visualizations in Appendix C).

In summary, our counterfactual ensemble explanations effectively contain diverse perturbations of the input time series. These perturbations trigger a change in the detected label of the anomalous sub-sequence, with a small number of altered features. The three approaches, ICE, DPE and FS, bring different insights on the model's prediction, the time series distribution and the possible perturbations to apply to change the former. Their relative advantages may therefore depend on the particular time series context and usage of the counterfactual explanation.

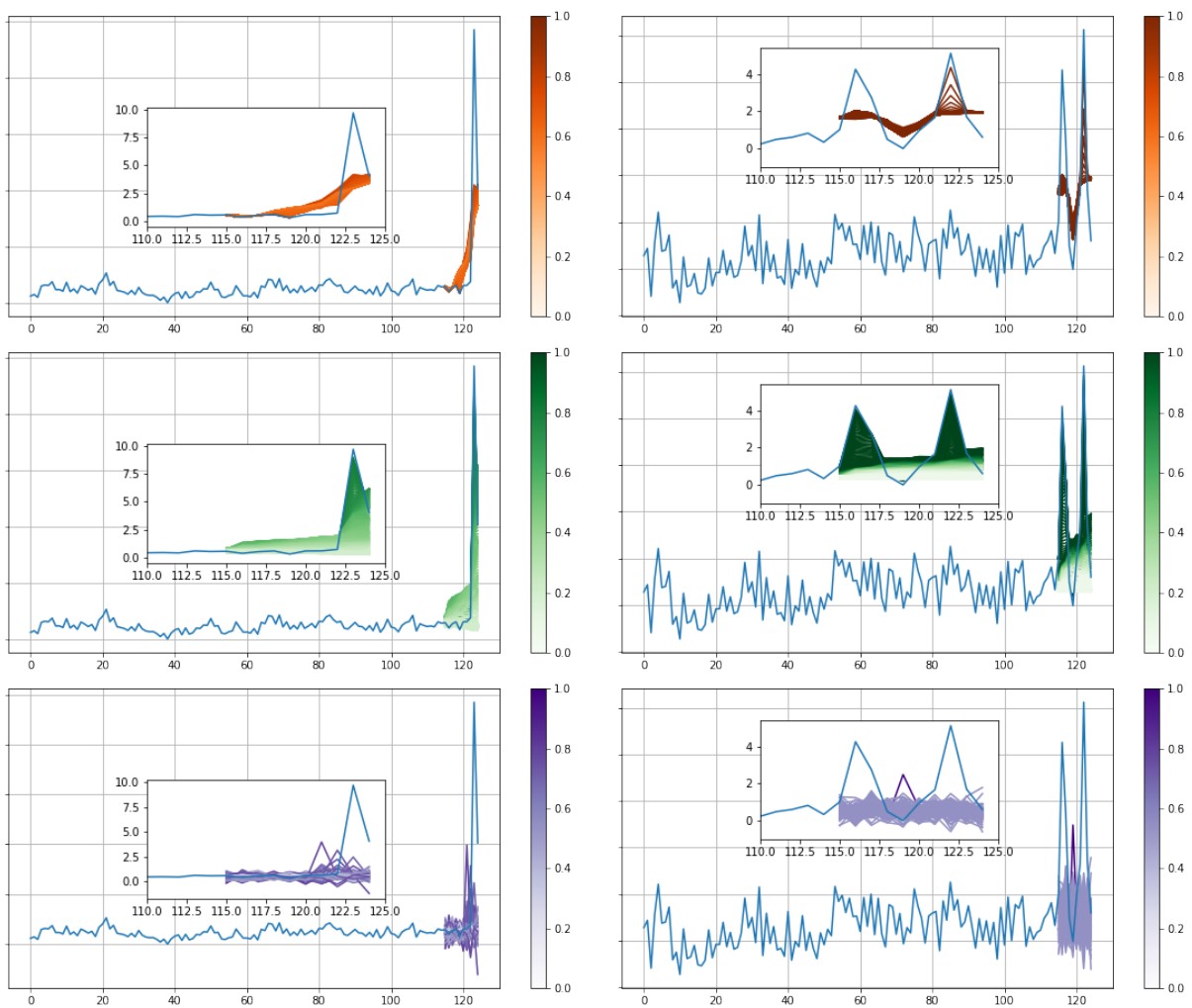

Figure 2: Time series windows containing an anomaly and our counterfactual ensemble explanations, obtained with DPE (first row), ICE (second row) and FS (third row) from the KPI dataset. The first (resp. second) column corresponds to an anomaly that has been detected by NCAD (resp. USAD). Each window includes a context part of 115 time stamps and an abnormal part of 10 time stamps at the end of the window. The original observations are plotted in blue, while the counterfactual examples appear in red, green or purple color scales for respectively DPE, ICE and FS.

| Dataset | Dimensions | Number of time series | Total number of time stamps | Total number of anomalies in test set |
|---------|-----------|----------------------|-----------------------------|----------------------------------------|
| KPI | 1 | 29 | 5922913 | 54560 |
| Yahoo | 1 | 367 | 609666 | 2963 |
| SMD | 38 | 28 | 1416825 | 29444 |
| SMAP | 25 | 55 | 584860 | 57079 |

Table 1: Succinct description of the four benchmark datasets

## 5.5 Numerical evaluation

The numerical results discussed in this section are obtained in the set-up described in Section 5.1. However, for parsimony of exposition, our results on the KPI and the SMAP datasets have been moved to Appendix B. We also add a partial sensitivity analysis of our method in Appendix E.

The results on univariate datasets (see Table 2 and Table 5 in Appendix B.1), show that our method has fairly small failure rates (except for the Yahoo dataset and the USAD model). In particular a rate smaller than 10% can be achieved with at least one variant in most pairs (model, dataset), leading to a consequent improvement over the naive procedure. We note that while the DPE variant seems to be valid more often than ICE on the NCAD model, it is the contrary for the USAD model; this difference is possibly due to the distinct internal mechanisms of these models.

Moreover, the analysis of the other explainability metrics supports the qualititative interpretation from Section 5.4. The Distance metric confirms than the gradient-based approaches, DPE and ICE, provides in almost all cases the closest counterfactuals in average, i.e., the least perturbed examples. Note that it sometimes occurs that the naive baseline has a small distance, however it always have a high failure rate. Besides, the Implausibility metrics validate the observation that FS generates the most realistic counterfactual examples in average, in particular in terms of Implausibility 1 (distance to median forecast sample) and Implausibility 3 (NLL under the probabilistic forecasting distribution). This is in fact quite expected since these quantities are directly derived from the forecasting sampling scheme. However, these counterfactuals are less smooth (higher score in Implausibility 2) than for DPE and ICE, which regularize the time series smoothness in the objective functions equation 2 and equation 1.

Finally, DPE and ICE provide a more diverse counterfactual ensemble in most cases in general, but their relative ranking is not clear from these experiments. We conjecture that this metric is particularly sensitive to the learning rate of the SGD algorithm, and the subsampling procedure after the objective minimization (see Section 4.1). In Appendix E, we test our first hypothesis on a small sample of anomalies. We observe in this case that the Diversity criterion is consistently higher for ICE, and greatly increases with the learning rate, at the cost of a higher failure rate.

The numerical results on the multivariate data sets are reported in Table 3 and Table 6 in Appendix B.1. These experiments showcase that our method also generates valid counterfactual ensemble explanations in this setting, with even a failure rate of 0% for the USAD model. Our method fails more frequently on the NCAD model, however, the sparse variants are more often successful. This indicates that imposing a sparsity constraint over the modified dimensions also helps to find valid counterfactuals. Consistent with the univariate datasets, FS produces the most realistic counterfactual examples while the gradient-based approach achieves a better Distance score. We note that in this case the Implausibility 3 metric is not available since the forecast distribution likelihood function in the DeepVAR model is not available [6]. Moreover, the sparse variants seem to correctly identify some of the anomalous channels (precision greater than 0.6 for the USAD model).

Nonetheless, we noted the greater difficulty of tuning the hyperparameters of our method and ranking its variants on these high-dimensional datasets compared to univariate data. In the latter, the default set of

---

[6]`https://ts.gluon.ai/stable/api/gluonts/gluonts.model.deepvar.html?highlight=deepvar#module-gluonts.model.deepvar` (accessed on September 11th 2022)

| **NCAD** on Yahoo | | | | | |
|---|---|---|---|---|---|
| Method | Failures (%) | Distance | Implausibility 1 | Implausibility 2 | Implausibility 3 | Diversity |
| DPE | **9.2** | 2.49 (4.91) | 1.23 (1.37) | 1.42 (2.19) | 2.21 (4.76) | 0.01 |
| ICE | 17.4 | **1.54 (1.21)** | 0.78 (1.37) | 2.26 (1.67) | 1.40 (5.17) | 0.05 |
| FS | 56.6 | 6.06 (16.38) | **0.27 (0.22)** | 3.36 (1.99) | **-0.29 (0.79)** | **0.10** |
| Naive | 72.2 | 2.69 (5.29) | 1.04 (1.26) | **1.32 (1.74)** | 1.89 (3.34) | 0.05 |

| **USAD** on Yahoo | | | | | |
|---|---|---|---|---|---|
| Method | Failures (%) | Distance | Implausibility 1 | Implausibility 2 | Implausibility 3 | Diversity |
| DPE | 29.1 | 5.20 (18.00) | 6.42 (26.81) | 0.42 (2.00) | 3.74 (6.96) | 0.05 |
| ICE | **25.5** | 6.66 (25.54) | 2.68 (11.46) | **0.40 (1.16)** | 2.48 (4.61) | **3.23** |
| FS | 65.1 | 14.48 (46.03) | **0.48 (0.72)** | 0.55 (0.58) | **-0.11 (1.12)** | 0.61 |
| Naive | 45.8 | **4.82 (18.36)** | 2.85 (16.31) | 0.52 (1.60) | 3.25 (6.00) | 3.19 |

Table 2: Performance of our explainability method and the naive baseline in terms of Validity, Closeness, Plausibility and Diversity on the Yahoo dataset and the NCAD (first panel) and USAD (second panel) anomaly detection models. We report the average scores and standard deviations (in brackets) over the counterfactual ensembles. We recall that *Implausibility 1* is the DTW distance to the median forecasting sample, *Implausibility 2* is the temporal smoothness, and *Implausibility 3* is the negative log-likelihood under the probabilistic forecasting output distribution. For all metrics except *Diversity*, we assume that a lower value is better, and the best score is highlighted in bold.

hyperparameters achieves an acceptable performance and allows to quickly compare the relative advantages of an approach for a specific pair (detection model, dataset). We therefore conclude by recalling that example-based explainability methods for multivariate time series are still in their early development, and providing general methods and tuning procedures to generate useful explanations over the instances of a dataset is still an open problem.

## 6 Discussion & Conclusion

In this work, we have introduced a novel type of post-hoc explainability method called *Counterfactual Ensemble Explanation* for anomaly detection models in time series. Our approach is model-agnostic, can be applied to any differentiable detection model, and is delineated into different variants according to the context. With DPE, one can apply a domain-specific perturbation mechanism to the input time series, while ICE does not require such specification. For high-dimensional time series, our sparse variants, *Sparse DPE* and *Sparse ICE* provide counterfactual examples modifying only a few dimensions of the time series. Additionally, we have proposed a gradient-free approach that uses a probabilistic forecasting technique as a generative scheme and can be applied to any detection model.

Our real-world experiments on four benchmark data sets show that the counterfactual framework, augmented with an ensemble approach, improves the interpretability of two deep-learning models and the anomalies the latter detects. In particular, our visualization tool allows to gauge the change in anomaly scores with respect to a large perturbation range of time series features. In the absence of competitive methods, we quantitatively compare our explanations to a *naive* counterfactual ensemble method using several explainability metrics.

In comparison to existing model-agnostic explainability methods for time series, our approach conveys more quantitative information on the model's sensitivity that a feature-saliency approach such as DynaMask (Crabbe & van der Schaar, 2021) and a richer contrastive explanation than single-counterfactual methods such as Delaney et al. (2021); Ates et al. (2021). Nonetheless, our proposed counterfactual ensemble expla-

| | | | **NCAD** on SMD | | | |
|---|---|---|---|---|---|---|
| Method | Failures (%) | Precision / Recall | Distance | Implausibility 1 | Implausibility 2 | Diversity |
| DPE | **17.1** | - | **8.46 (13.07)** | 50.76 (110.40) | 12.12 (28.13) | 1.21 |
| ICE | 42.9 | - | 79.62 (120.27) | 23.69 (28.24) | 47.73 (58.32) | **4639.11** |
| Sparse DPE | 20.0 | **0.22** / 0.10 | 36.12 (77.87) | 29.03 (54.36) | 5.61 (11.30) | **4687.05** |
| Sparse ICE | 20.0 | 0.20 / **0.33** | 26.01 (37.07) | 62.65 (107.25) | 10.88 (16.72) | 174.39 |
| FS | 30.0 | - | 78.46 (157.57) | **1.62 (2.33)** | **1.49 (1.31)** | 35.88 |
| Naive | 79.8 | - | 25.06 (53.66) | 45.45 (93.03) | 9.42 (18.79) | 3255.92 |
| | | | **USAD** on SMD | | | |
| Method | Failures (%) | Precision / Recall | Distance | Implausibility 1 | Implausibility 2 | Diversity |
| DPE | **0.0** | - | 139.02 (261.44) | 258.31 (464.18) | 41.19 (79.11) | **23339.20** |
| ICE | **0.0** | - | **31.81 (9.27)** | 342.19 (708.88) | 22.64 (45.16) | 0.52 |
| Sparse DPE | **0.0** | **0.68** / 0.07 | 115.48 (206.46) | 293.70 (679.88) | 19.84 (34.98) | 105.45 |
| Sparse ICE | **0.0** | 0.61 / **0.28** | 216.44 (316.05) | 172.58 (475.15) | **8.35 (17.52)** | 477.43 |
| FS | **0.0** | - | 366.57 (672.44) | **18.10 (48.25)** | **8.57 (20.63)** | 12175.65 |
| Naive | 73.4 | - | 49.83 (60.13) | 475.42 (879.38) | 27.45 (45.43) | 649.44 |

Table 3: Performance of our explainability method and the naive baseline in terms of Validity, Closeness, Plausibility and Diversity on the SMD dataset and the NCAD (first panel) and USAD (second panel) anomaly detection models. We report the average scores and standard deviations (in brackets) over the counterfactual ensemble. We recall that *Implausibility 1* is the DTW distance to the median forecasting sample and *Implausibility 2* is the temporal smoothness. For all metrics except *Diversity*, *Precision* and *Recall*, we assume that a lower value is better, and the best score is highlighted in bold.

nation for time series models is an attempt in the interpretation of these models using diverse instance-based methods, in particular in the challenging high-dimensional context.

Although our method offers greater flexibility, better explainability performances and specific interpretation might be achieved if more assumptions are put on the detection model. In particular, similarly to Rodriguez et al. (2021), we could adapt our gradient-based approach to use the internal representations of the model rather than the raw time series. Moreover, aggregating the information contained in diverse explanations is still an open problem. One possible extension of our ensemble method would be to provide a rank over the counterfactual examples according to a utility or feasibility metric.

### Broader Impact Statement

We do not see any direct negative impact of our work, however ethical concerns could come from the type of time series data our methodology is applied to. Moreover, our method does not rank the counterfactuals in the set of solutions, and their acceptability needs to be assessed by a domain expert. An extension of our method could be to include fairness constraints in the optimisation objective to obtain "fair" counterfactuals.

### Acknowledgments

We thank Chris Russell, Dominik Janzig, Lenon Minorics and Lorenzo Stella for their insightful comments on our work.

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

## A    Technical details and performance of the selected anomaly detection models

In this section, we provide some technical details on the two anomaly detection models selected for the evaluation of our explainability method reported in Section 5. In Table 4, we report their anomaly detection performance on the benchmark datasets, after training with the hyperparameter sets reported in their respective papers when available. Otherwise, we select the models' hyperparameters on a validation set (20% of the time series) using the best adjusted F1-score.

**Neural Contextual Anomaly Detection (NCAD) Carmona et al. (2021) :** This method splits time series into subwindows $(W^i)_i$ and embeds them using a temporal convolutional network (TCN). Each $W^i$ is subdivided into a context part and a suspect part (typically much smaller than the former), i.e., $W^i = [W_C^i, W_S^i]$. An embedding of the context window $W_C^i$ is also computed by the TCN, then the distance between the embeddings of $W_i$, denoted $z^i$, and $W_C^i$, denoted $z_C^i$, is evaluated. The algorithm finally labels $W_S^i$ as anomalous if the latter distance is greater than a chosen threshold, i.e, if $d(z^i, z_C^i) > \eta$ with $d(.,.)$ the Euclidean distance for instance and $\eta > 0$. The intuition behind this method is that a large distance between the embeddings of a window and its context part means that the suspect part induces a significant shift of $z_C^i$ in the embedding space. Since the embedding of the context window should reflect the normal behaviour, this deviation thus indicates the presence of an anomaly in $W_S^i$. For our experiments, we use the open-source implementation. [7]

**UnSupervised Anomaly Detection (USAD) Audibert et al. (2020):** This reconstruction model splits time series into subwindows that are reconstructed by a LSTM-based AutoEncoder. The latter contains a neural network, called encoder, that embeds each window into a latent representation, and another neural network, called decoder, that maps back the embedding into the original input space. The reconstruction error, i.e., the distance in the time series domain between the original input and the reconstructed output, is used as an anomaly score (a high value of this error leads to the corresponding window to be labelled as anomalous). We use the open source implementation provided by the authors [8] and the hyperparameters provided in the paper for the two multivariate data sets, i.e. SMD and SMAP. For the KPI dataset, the final USAD model is trained for 80 epochs and has windows of size 5, hidden size of 10 and downsampling rate of 0.01. For the Yahoo data, the window size is 10, hidden size of 10 and downsampling rate of 0.05.

| Model | KPI | Yahoo | SMD | SMAP |
|-------|-----|-------|-----|------|
| NCAD | 0.789 | 0.772 | 0.806 | 0.922 |
| USAD | 0.946 | 0.741 | 0.643 | 0.972 |

Table 4: F1-scores of the two anomaly detection models, i.e., NCAD and USAD, on the four benchmark datasets.

---

[7]https://github.com/Francois-Aubet/gluon-ts/tree/adding_ncad_to_nursery/src/gluonts/nursery/ncad
[8]https://curiousily.com/posts/time-series-anomaly-detection-using-lstm-autoencoder-with-pytorch-in-python/

| NCAD on KPI | | | | | | |
|---|---|---|---|---|---|---|
| Method | Failures (%) | Distance | Implausibility 1 | Implausibility 2 | Implausibility 3 | Diversity |
| DPE | **3.9** | 5.94 (15.78) | 2.16 (4.71) | 3.21 (29.62) | 2.74 (2.57) | **1.18** |
| ICE | 19.6 | **3.08 (1.21)** | 15.31 (115.12) | 31.67 (206.70) | 2.07 (2.06) | 0.26 |
| FS | 6.0 | 32.05 (173.76) | **0.21 (0.20)** | **2.42 (1.97)** | **-0.56 (1.14)** | 0.12 |
| Naive | 53.4 | 11.82 (74.03) | 2.90 (4.49) | 4.57 (7.64) | 3.48 (2.51) | 0.54 |
| USAD on KPI | | | | | | |
| Method | Failures (%) | Distance | Implausibility 1 | Implausibility 2 | Implausibility 3 | Diversity |
| DPE | 5.0 | 25.22 (121.60) | 9.40 (65.02) | 1.03 (8.16) | 3.13 (3.47) | 13.10 |
| ICE | **3.5** | **6.52 (7.30)** | 4.99 (63.31) | 0.50 (4.43) | 1.27 (1.98) | 0.28 |
| FS | 6.8 | 38.56 (189.88) | **0.33 (0.29)** | **0.38 (0.28)** | **-0.08 (1.12)** | 0.26 |
| Naive | 45.4 | 31.93 (154.62) | 2.77 (3.93) | 1.42 (6.01) | 2.81 (2.48) | **69.88** |

Table 5: Performance of our explainability method and the naive baseline in terms of Validity, Closeness, Plausibility and Diversity on the KPI dataset and the NCAD (first panel) and USAD (second panel) anomaly detection models. We report the average scores and standard deviations (in brackets) over the counterfactual ensemble. We recall that *Implausibility 1* is the DTW distance to the median forecasting sample, *Implausibility 2* is the temporal smoothness, and *Implausibility 3* is the negative log-likelihood under the probabilistic forecasting output distribution. For all metrics except *Diversity*, we assume that a lower value is better, and the best score is highlighted in bold.

## B    Additional numerical results

In this section, we report quantitative evaluations of our explainability method that could not be included in the main text due to space limitation. This section notably contains the results on two benchmark datasets using the procedure described in Section 5, and an additional analysis on False Positives.

### B.1    Numerical evaluation on the KPI and SMAP datasets

The results on the KPI and SMAP dataset are respectively in Table 5 and Table 6. Note that these results are included in the discussion in Section 5.5.

### B.2    Numerical evaluation on False Positives

In the practical use of anomaly detection models, explanations can also be needed when the model wrongly detects an anomaly in a time series. We recall that we call False Positives the anomalies detected by the model that are not ground-truth anomalies. We present here a numerical evaluation on the False Positives detected by NCAD in the KPI benchmarck dataset. The results in Table 7 can be compared to the results obtained on True Positives (i.e., the ground-truth, detected anomalies) reported in the first panel of Table 5 . We observe that in this case ICE achieves 0% failure rate (instead of almost 20 %), and the naive method has also a significantly smaller number of failures. Moreover, all methods seem to perform better in terms of the Distance and Implausibility metrics. This is probably due to the fact that False Positives need less perturbation to become not anomalous for the model, e.g. if they lie close to the model's local decision boundary. Therefore they may inherently be less distant to the normal behaviour than True Positives and

| NCAD | | | | | |
|---|---|---|---|---|---|
| Method | Failures (%) | Diversity | Distance | Implausibility 1 | Implausibility 2 |
| DPE | 41.7 | 0.002 | 0.19 (0.40) | 0.21 (0.28) | **0.01 (0.03)** |
| DPE sparse | 27.8 | 0.004 | 0.22 (0.42) | 0.29 (0.38) | 0.03 (0.04) |
| ICE | **5.6** | **0.067** | 0.26 (0.14) | 0.39 (0.37) | 0.15 (0.09) |
| ICE sparse | 23.6 | 0.016 | 0.15 (0.08) | 0.22 (0.21) | 0.09 (0.06) |
| FS | 87.6 | 0.012 | 0.56 (0.77) | **0.05 (0.04)** | 0.05 (0.04) |
| Naive | 84.5 | 0.003 | **0.06 (0.08)** | 0.09 (0.03) | 0.02 (0.03) |
| USAD | | | | | |
| Method | Failures (%) | Diversity | Distance | Implausibility 1 | Implausibility 2 |
| DPE | **0.0** | 0.02 | 0.62 (0.65) | 0.96 (0.53) | 0.06 (0.05) |
| DPE sparse | **0.0** | 0.02 | 0.78 (0.85) | 0.82 (0.50) | 0.05 (0.06) |
| ICE | **0.0** | 0.17 | 0.74 (0.75) | 0.87 (0.43) | **0.04 (0.03)** |
| ICE sparse | **0.0** | **0.18** | 0.72 (0.76) | 0.88 (0.42) | 0.06 (0.02) |
| FS | 56.8 | 0.02 | 2.23 (1.14) | **0.09 (0.01)** | 0.10 (0.03) |
| Naive | 46.9 | 0.01 | **0.14 (0.04)** | 0.23 (0.02) | 0.07 (0.02) |

Table 6: Performance of our explainability method and the naive baseline in terms of Validity, Closeness, Plausibility and Diversity on the SMAP dataset and the NCAD (first panel) and USAD (second panel) anomaly detection models. We report the average scores and standard deviations (in brackets) over the counterfactual ensemble. We recall that *Implausibility 1* is the DTW distance to the median forecasting sample and *Implausibility 2* is the temporal smoothness. For all metrics except *Diversity*, *Precision* and *Recall*, we assume that a lower value is better, and the best score is highlighted in bold.

thus easier instances for our counterfactual explanation method. Besides, the Diversity metric is smaller for DPE and ICE, likely as another effect of the smaller amount of perturbation needed.

| NCAD | | | | | | |
|---|---|---|---|---|---|---|
| Method | Failures (%) | Distance | Implausibility 1 | Implausibility 2 | Implausibility 3 | Diversity |
| DPE | 8.8 | **2.22 (1.87)** | 2.44 (2.50) | 2.36 (2.05) | 2.15 (2.22) | 0.02 |
| ICE | **0.0** | 4.36 (3.00) | **0.28 (0.45)** | **0.61 (0.34)** | 0.22 (0.93) | 0.12 |
| FS | 6.6 | 4.17 (2.91) | 0.30 (0.31) | 3.54 (3.61) | **-0.22 (0.95)** | **0.43** |
| Naive | 33.3 | 2.74 (2.22) | 2.19 (2.43) | 2.97 (2.56) | 2.79 (2.29) | 0.16 |

Table 7: Performance of our explainability method and the naive baseline in terms of Validity, Closeness, Plausibility and Diversity on the false positives in the KPI data detected by the NCAD model. We report the average scores and standard deviations (in brackets) over the counterfactual ensemble. We recall that *Implausibility 1* is the DTW distance to the median forecasting sample, *Implausibility 2* is the temporal smoothness, and *Implausibility 3* is the negative log-likelihood under the probabilistic forecasting output distribution. For all metrics except *Diversity*, we assume that a lower value is better, and the best score is highlighted in bold.

## C  Complementary visualizations of the explanations

In this section, we report additional visualizations of our counterfactual explanations, as well as illustrations of the sparsity induced by the sparse variants of DPE and ICE. Figures 3 and 4 are visualizations applied to the univariate datasets and respectively the NCAD and USAD. The advantage of Sparse ICE compared to the plain version ICE is shown in Figure 5, where only four channels of the multi-dimensional time series window are plotted. For this anomaly, only one of these dimensions contains an anomalous observation

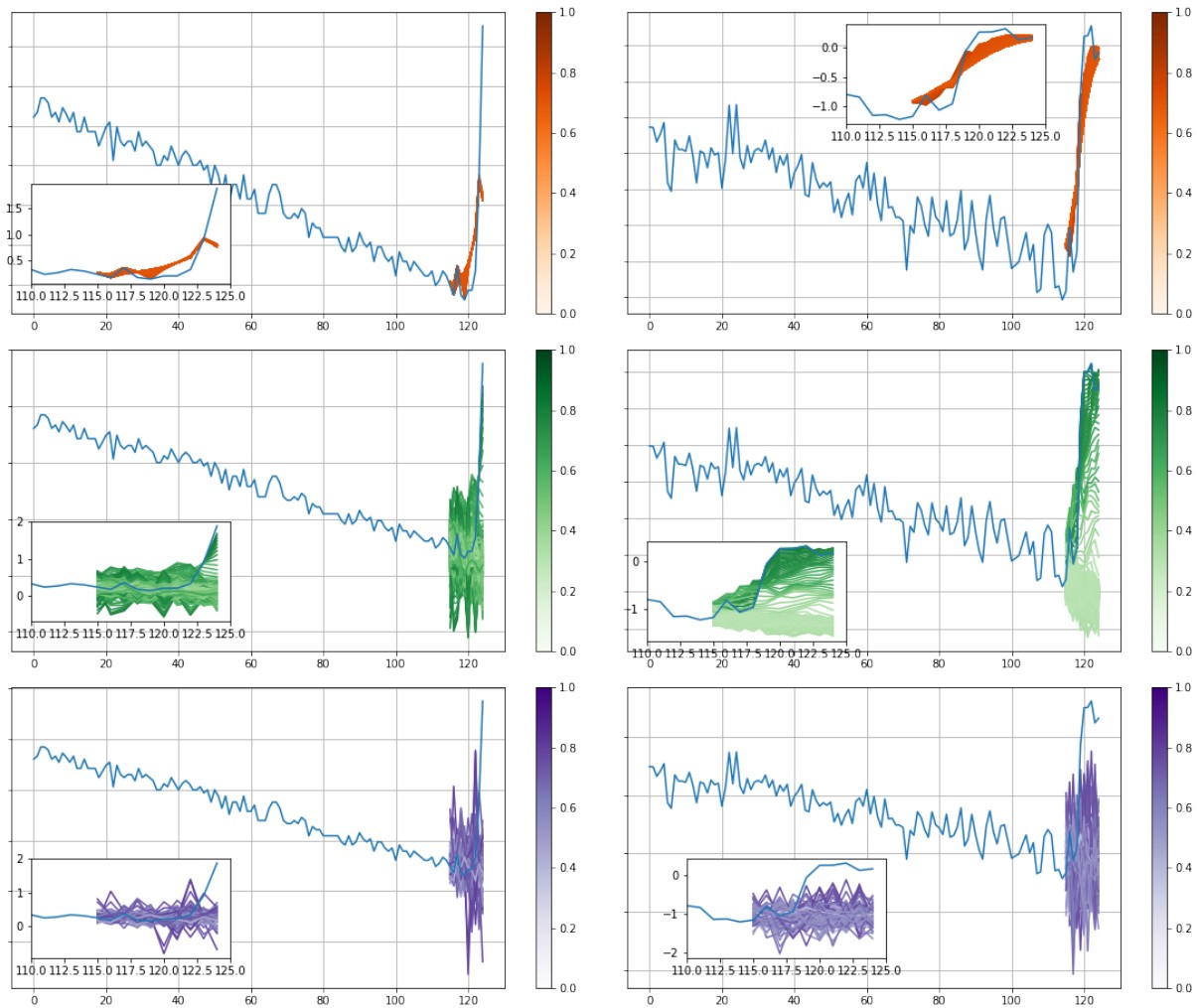

Figure 3: Anomalous windows and counterfactual ensemble explanations obtained with DPE (first row), ICE (second row) and FS (third row) on anomalies in the KPI data set detected by the NCAD model. The columns correspond to two different anomalies. The windows include a context part of 115 time stamps and an abnormal part of 10 time stamps. The original sub-sequence is plotted in blue, while the explanations are in red, green or purple colors for the different variants.

but the counterfactual explanation obtained with the plain ICE perturbs four of them. In contrast, the Sparse ICE variant keeps two dimensions without anomalous features unchanged, leading to a more accurate and readable explanation on this particular anomaly. Similarly, Figure 6 shows two perturbation maps corresponding to examples generated by DPE and its sparse variant. While the plain DPE produces *globally* sparse maps (i.e., in the temporal and dimensional features), Sparse DPE is sparse in dimensions, leading to perturbed examples with few modified channels.

## D    Illustration of the hyperparameters selection

In this section, we illustrate the hyperparameters selection procedure for our gradient-based method. For each dataset and model, we run our algorithm with several configurations as described in Section 5.3 and select the final one using the failure rate and the Implausibility 1 metric. More precisely, we select a threshold of acceptable failure rate (e.g., 10% or 20%), then amongst the configurations achieving a lower value of the latter, we select the one with the lowest Implausibility 1 value. Figures 7,  8,  10 and  9 show the values of

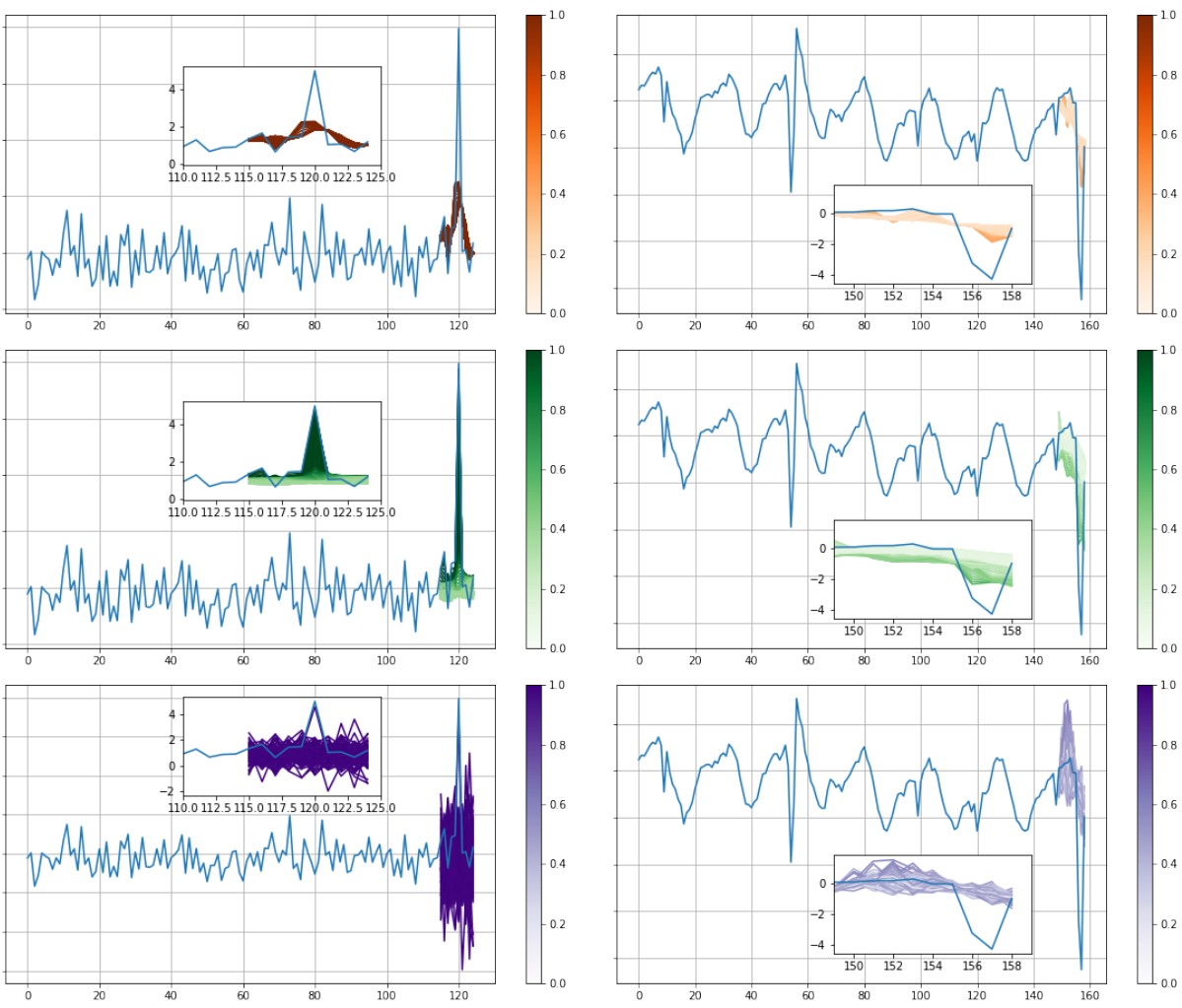

Figure 4: Anomalous windows and counterfactual ensemble explanations obtained with DPE (first row), ICE (second row) and FS (third row) on anomalies in the KPI and Yahoo data sets detected by the USAD model. The rows correspond to different anomalies. The windows include a context part of 115 timestamps and an abnormal part of 10 timestamps. The original subsequence is plotted in blue, while the explanations are in red, green or purple colors for the different variants.

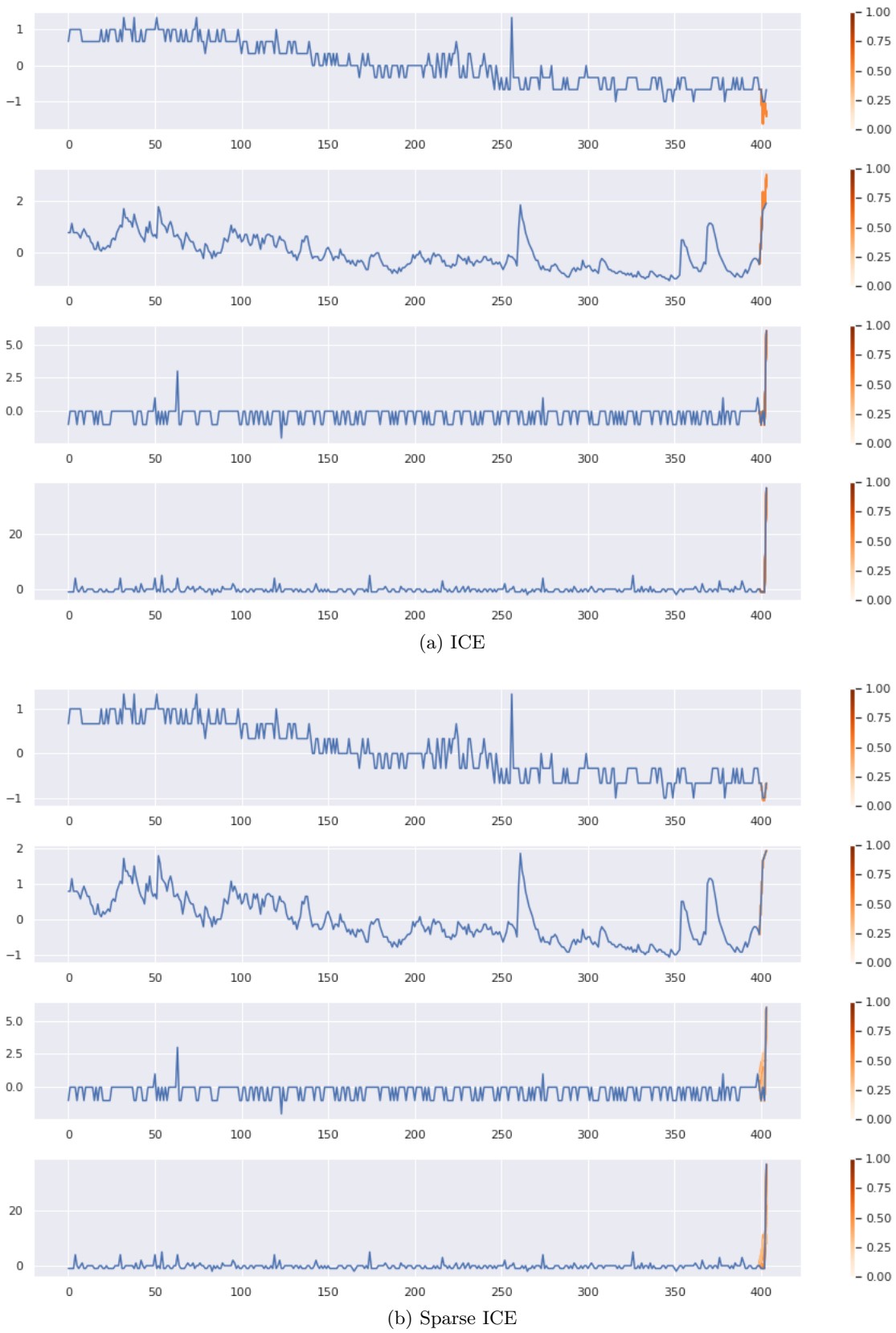

(a) ICE

(b) Sparse ICE

Figure 5: Counterfactual explanation obtained with ICE (a) and the sparse variant (b). The different rows correspond respectively to the first, third, ninth and twelfth dimensions of a subsequence in the SMD dataset. Amongst them, only the fourth two (twelfth dimension) contains an anomalous observation in the last timestamp of the displayed window, detected by the NCAD model. While ICE (a) modifies all the plotted dimensions, Sparse ICE only perturbs the third and fourth (i.e., the ninth and twelfth dimension).

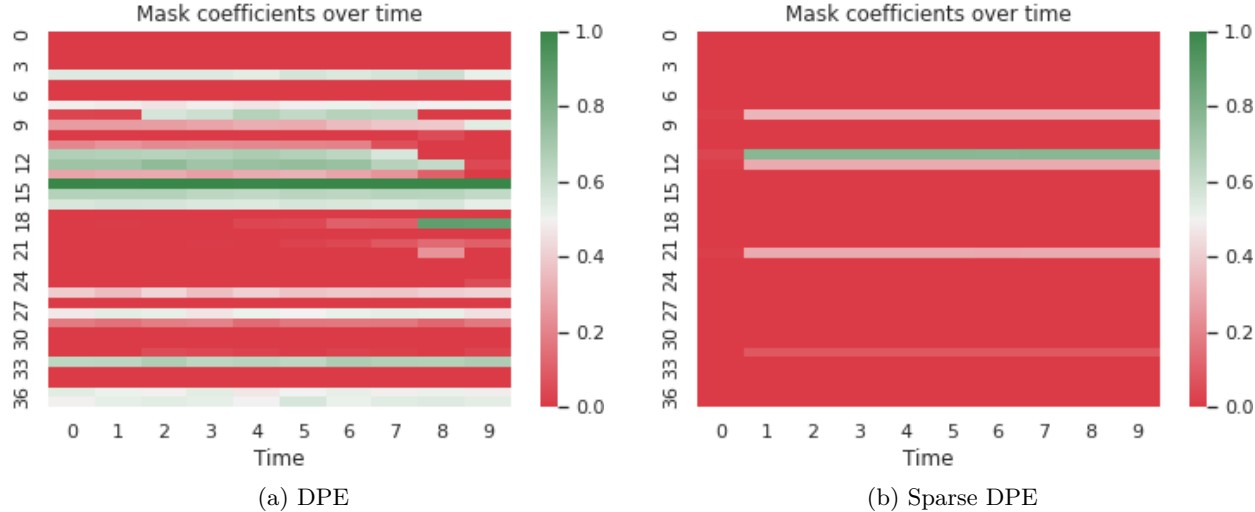

|  (a) DPE | (b) Sparse DPE |

Figure 6: Perturbation maps of counterfactual examples in the explanations generated by DPE (a) and its sparse variant (b) on one anomaly in the SMD dataset detected by NCAD. We recall that the rows of each mask correspond to the different dimensions of the time series and the columns to the successive timestamps in the suspect window (see Section 4). The color bars on the right sides of the maps indicate the values (between 0 and 1) of these maps along the time series features.

| Dataset | Model type | Number of layers | Hidden size | training epochs | learning rate | prediction length |
|---------|-----------|-----------------|-------------|-----------------|---------------|-------------------|
| KPI | FFNN | 1 | 32 | 100 | 0.001 | 10 |
| Yahoo | FFNN | 1 | 32 | 100 | 0.001 | 10 |
| SMD | DeepVAR | 4 | 40 | 150 | 0.001 | 10 |
| SWaT | DeepVAR | 4 | 40 | 150 | 0.001 | 10 |

Table 8: Hyperparameters of the Probabilistic Forecasting models used in the gradient-free approach on the four benchmark datasets.

these metrics for all explored configurations for each model and dataset. Lastly, in Tables 9, 10, 11 and 12, we report the selected configurations for respectively DPE, ICE, Sparse DPE and Sparse ICE on the benchmark datasets. Besides, the hyperparameters of the gradient-free approach can be found in Table 8.

| Dataset | Perturbation | $\sigma_{max}$ | learning rate | $\lambda_2$ | $\lambda_T$ |
|---------|-------------|----------------|---------------|-------------|-------------|
| NCAD-KPI | Gaussian blur | 3.0 | 0.01 | 0.01 | 0.1 |
| NCAD-Yahoo | Gaussian blur | 10.0 | 0.01 | 0.001 | 0.1 |
| NCAD-SMD | Gaussian blur | 20.0 | 0.01 | 0.0 | 1.0 |
| NCAD-SMAP | Gaussian blur | 10.0 | 0.01 | 1.0 | 1.0 |
| USAD-KPI | Gaussian blur | 3.0 | 0.01 | 0.001 | 1.0 |
| USAD-Yahoo | Gaussian blur | 10.0 | 0.01 | 0.001 | 1.0 |
| USAD-SMD | Gaussian blur | 20.0 | 0.1 | 0.001 | 0.01 |
| USAD-SMAP | Gaussian Blur | 20.0 | 0.01 | 0.01 | 0.1 |

Table 9: Hyperparameters of the DPE algorithm on the four benchmark datasets.

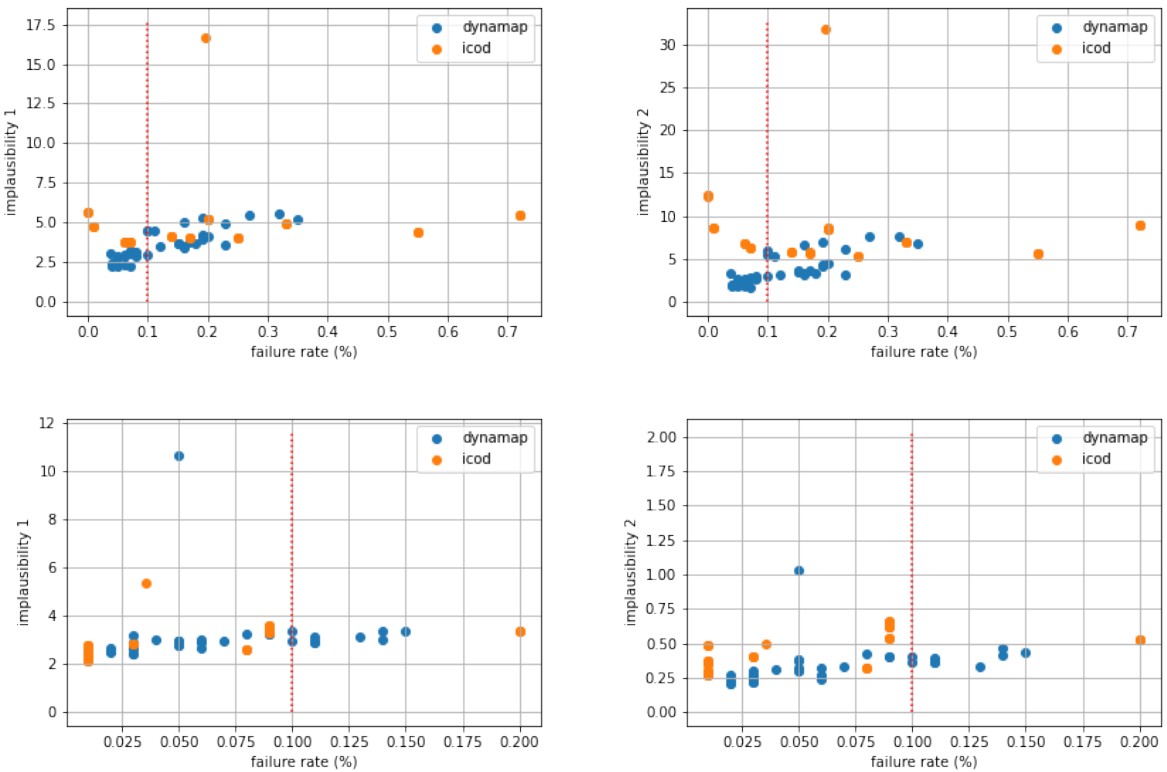

Figure 7: Implausibility measures 1 (left column) and 2 (right column) versus failures rates for different sets of hyperparameters of the ICE and DPE algorithms and their sparse variants applied to the NCAD (first row) and USAD (second row) models on a the KPI dataset. The metrics are computed over a validation set of 5 time series and the failure rate's threshold is 10% (red dotted line).

| Dataset | learning rate | $\lambda_1$ | $\lambda_2$ | $\lambda_T$ |
|---------|---------------|-------------|-------------|-------------|
| NCAD-KPI | 0.1 | 0.01 | 0.01 | 1.0 |
| NCAD-Yahoo | 0.1 | 0.01 | 0.01 | 1.0 |
| NCAD-SMD | 0.1 | 0.01 | 0.01 | 0.1 |
| NCAD-SMAP | 0.1 | 0.1 | 0.1 | 1.0 |
| USAD-KPI | 0.1 | 0.001 | 0.001 | 1.0 |
| USAD-Yahoo | 0.1 | 0.001 | 0.001 | 1.0 |
| USAD-SMD | 1000.0 | 0.01 | 0.01 | 1.0 |
| USAD-SMAP | 1.0 | 0.001 | 0.001 | 1.0 |

Table 10: Hyperparameters of the ICE algorithm on the four benchmark datasets.

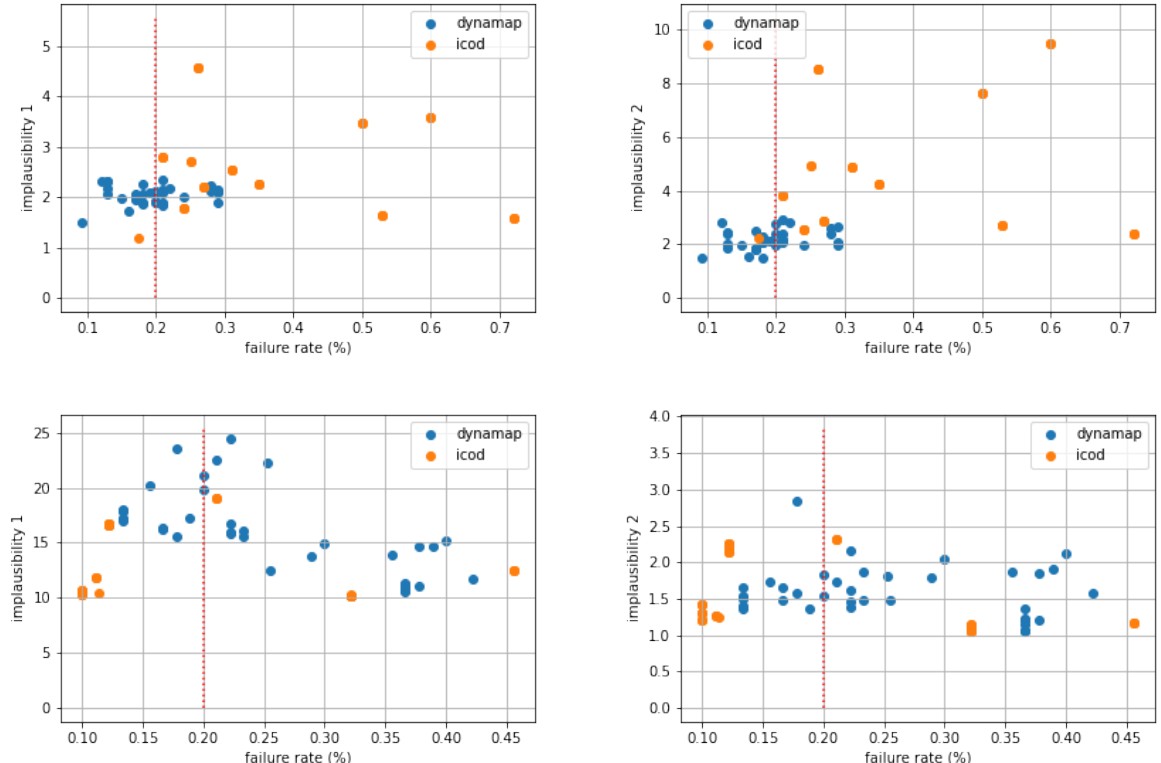

Figure 8: Implausibility measures 1 (left column) and 2 (right column) versus failures rates for different sets of hyperparameters of the ICE and DPE algorithms and their sparse variants applied to the NCAD (first row) and USAD (second row) models on a the Yahoo dataset. The metrics are computed over a validation set of 15 time series and the failure rate's threshold is 25% (red dotted line).

| Dataset | Perturbation | $\sigma_{max}$ | learning rate | $\lambda_1$ | $\lambda_2$ | $\lambda_T$ |
|---|---|---|---|---|---|---|
| NCAD-SMD | Gaussian blur | 20.0 | 0.1 | 0.01 | 0.01 | 0.1 |
| NCAD-SMAP | Gaussian Blur | 10.0 | 0.01 | 0.1 | 0.1 | 0.1 |
| USAD-SMD | Gaussian Blur | 20.0 | 0.01 | 0.01 | 0.01 | 1.0 |
| USAD-SMAP | Gaussian Blur | 20.0 | 0.1 | 0.01 | 0.01 | 0.1 |

Table 11: Hyperparameters of the Sparse DPE algorithm on the two benchmark multivariate datasets.

| Dataset | learning rate | $\lambda_1$ | $\lambda_2$ | $\lambda_T$ |
|---|---|---|---|---|
| NCAD-SMD | 0.1 | 0.01 | 0.01 | 0.1 |
| NCAD-SMAP | 0.1 | 0.1 | 0.1 | 1.0 |
| USAD-SMD | 10000.0 | 0.01 | 0.01 | 0.1 |
| USAD-SMAP | 1.0 | 0.001 | 0.001 | 1.0 |

Table 12: Hyperparameters of the Sparse ICE algorithm on the two benchmark multivariate datasets.

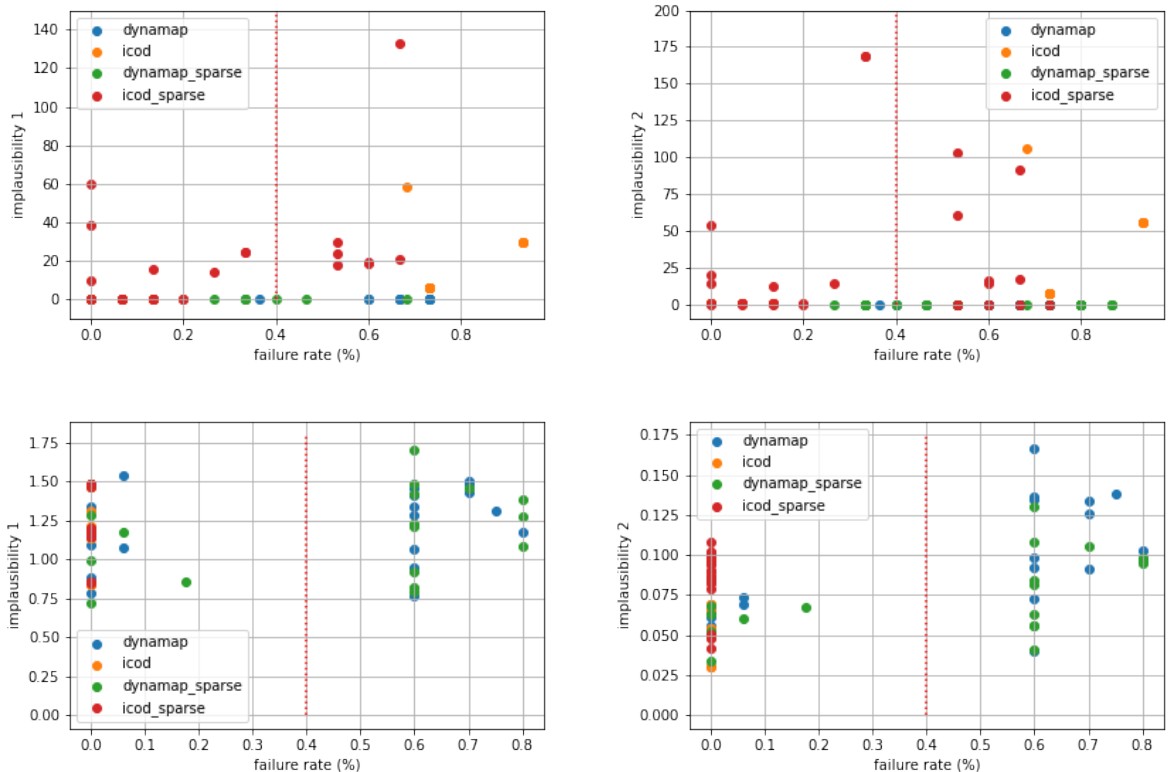

Figure 9: Implausibility measures 1 (left column) and 2 (right column) versus failures rates for different sets of hyperparameters of the ICE and DPE algorithms and their sparse variants applied to the NCAD (first row) and USAD (second row) models on a the SMAP dataset. The metrics are computed over a validation set of 40 time series and the failure rate's threshold is 25% (red dotted line).

| Variant | Perturbation | $\sigma_{max}$ | learning rate | $\lambda_1$ | $\lambda_2$ | $\lambda_T$ | $N$ |
|---|---|---|---|---|---|---|---|
| ICE | - | - | 0.1 | 0.01 | 0.01 | 0.01 | 100 |
| DPE | Gaussian Blur | 3.0 | 0.01 | - | 0.1 | 0.01 | 100 |

Table 13: Default set of hyperparameters for our gradient-based counterfactual ensemble method.

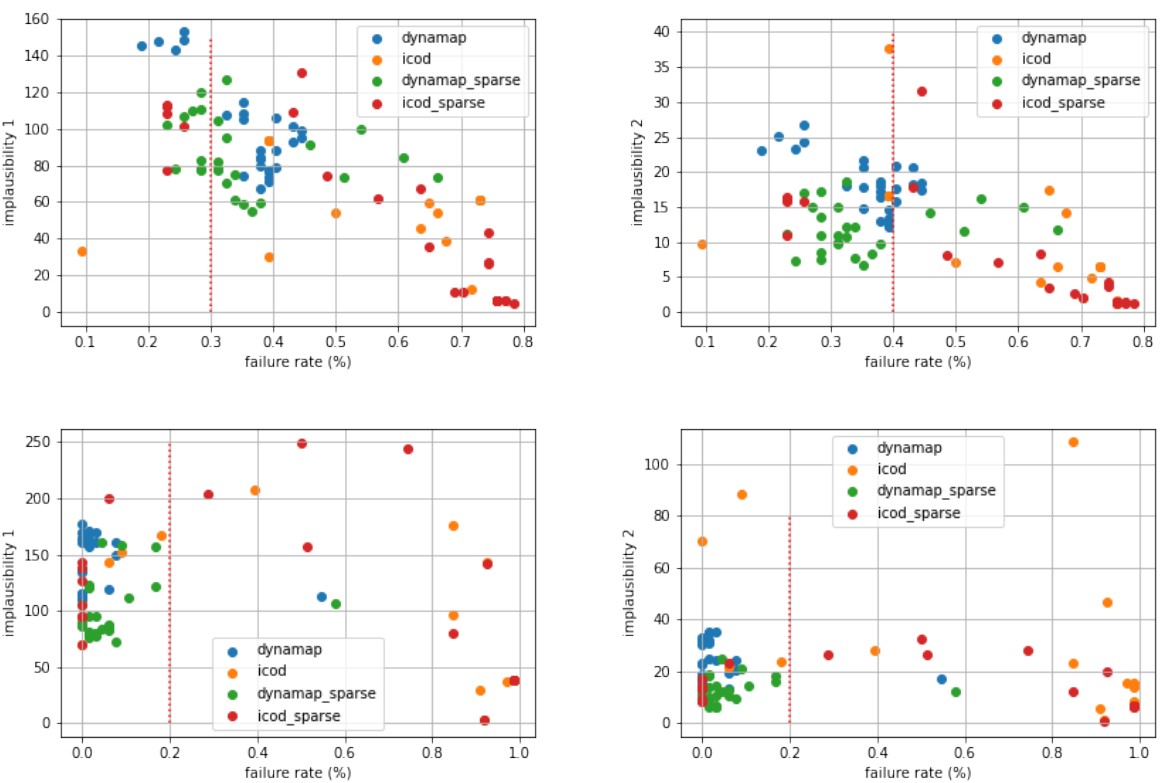

Figure 10: Implausibility measures 1 (left column) and 2 (right column) versus failures rates for different sets of hyperparameters of the ICE and DPE algorithms and their sparse variants applied to the NCAD (first row) and USAD (second row) models on a the SMD dataset. The metrics are computed over a validation set of 6 time series and the failure rate's threshold is 40% for NCAD and 20% for USAD (red dotted lines).

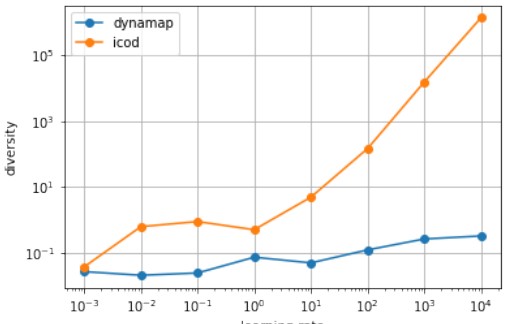 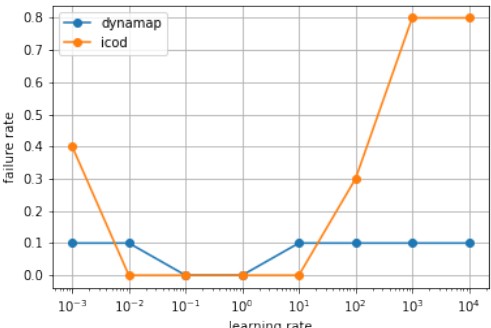

Figure 11: Diversity of the counterfactual ensemble (left) and failure rate of our counterfactual method (right) versus the learning rate of the SGD algorithm for the two variants of our method, ICE and DPE.

# E    Sensitivity of the Diversity criterion to the learning rate parameter

In this section we report a small-scale study of the influence of the learning rate in the SGD algorithm on the Diversity metric, in our gradient-based approach. We evaluate the latter metric on 10 anomalies detected by the NCAD model in the KPI dataset, obtained with DPE and ICE with learning rates in the set $\{0.001, 0.01, 0.1, 1, 10, 100, 1000, 10000\}$. The other hyperparameters of our method are the same as in Section 5.5. Figure 11 shows the evolution of the Diversity score (left panel) and failure rate (right panel) when the learning rate increases. We observe that the diversity is always higher for ICE than DPE, and dramatically increases when the learning rate is greater than 1 for the former. However, failure rate also skyrockets for high learning rates.

