# OpenReview forum: "Diverse Counterfactual Explanations for Anomaly Detection in Time Series"
_TMLR — Rejected by TMLR_

### Review · Reviewer_vjgL · 2022-11-16

**Summary Of Contributions:**

The paper studies anomaly detection in time series using non-transparent models which may require explanations for predictions made. To this end, the authors propose using explanations generated by finding so-called counterfactuals for model inputs which are new input examples that are a) close to the original example and b) predicted to be non-anomalous according to the model under study. To enhance interpretation, the authors suggest that such counterfactuals should be a diverse set, an ensemble, and propose a heuristic for identifying such a set. Two objectives for finding counterfactuals for differentiable models using gradient descent are proposed, as well as a method based on generative models. The approach is evaluated on benchmark tasks for anomaly detection by providing counterfactual explanations for the predictions made by two state-of-the-art models.

**Audience:**

Yes

**Broader Impact Concerns:**

None.

**Claims And Evidence:**

No

**Requested Changes:**

- Better align the claims/motivation with the results. For example, in the abstract, the authors claim to "show that our algorithm can produce valuable explanations". Valuable to whom? Valuable in what way? Additionally, they claim to "improve the interpretability of our explanation for high-dimensional time-series anomalies". This is not currently evaluated.
- Expand the study and discussion of methods for achieving diversity. As described above, the chosen method Is not well justified mathematically (there may be multiple local optima) or conceptually (it is not clear why humans would prefer this proposed kind of diversity). Only one baseline is included in the experiments, and it is neither of the methods described in 4.1.3. Is there a reason these could not be used in the empirical evaluation?
- Change the citation format to have parentheses around non-textual citations (most of them). It is currently very hard to read parts with many citations.
- In the introduction, the authors compare their setting to adversarial examples and claim that these are not quite relevant since they are weakly constrained and therefore implausible. This is incorrect, there is a large literature on adversarial examples, some of which go to great lengths to make adversarial examples plausible. Even without this in mind, I think the similarity warrants additional investigation: are there adversarial example methods that could be used as baselines?


**Strengths And Weaknesses:**

**Strengths**
- The paper has a clear structure and is generally very well written. It is easy to follow arguments and descriptions of methodological contributions.
- The related work section provides relevant context for the work.
- The empirical evaluation appears to be performed on relevant tasks and with relevant methods (I am not an expert in this anomaly detection, specifically).

**Weaknesses**
 - The paper follows a long line of work attempting to explain the predictions made by differentiable models by exploiting gradients with respect to the input for a given example. In this case, the gradients are used to tweak the example until it changes the prediction made by the model. It is not clear from the paper how these "counterfactuals" are intended to be used and, arguably, their quality cannot be assessed without specifying this. In the paper, explanations are given for three examples (Figure 1 & 2) but it is difficult to ascertain the practical value of the method from these examples since the anomalies and plausible counterfactuals are clear even without the model predictions. Moreover, there is no discussion about how to handle possibly erroneous predictions made by the model. Are explanations as useful when the anomaly detection model is wrong as when it is correct?
- The paper claims to make three methodological contributions, a) counterfactual ensemble explanations, b) sparse explanations, and c) visualization. In my opinion, each of these have strong limitations which are not discussed in the paper:

  a) The ensemble is created by storing example inputs on the optimization path of gradient descent minimizing the same objective. This is unlikely to create meaningful diversity. For the type of large differentiable models that the authors target in this work, it is unlikely that the objective has a unique local (or global) minimum, and hence, there may be many counterfactuals that are fundamentally different but do not lay on the same path of gradient descent (since this will explore only one local minimum). In 4.1.3, alternatives to this strategy are discussed, such as including a proximity penalty between examples or selecting optima for different random initializations. It is clear from the discussion that these are not just different strategies, but also that they pursue different *goals*. It is not clear from the paper in which scenario the proposed heuristic is preferable (other than for computational efficiency).

  b) Generally, the motivation for the sparsity penalty is that "humans prefer simpler explanations" (§4.2), and that this is useful in high-dimensional settings. However, no explanations for high-dimensional are illustrated in the paper, nor were such explanations evaluated with humans interpreting them.

  c) The proposed visualization scheme amounts to plotting the counterfactual examples. Although generally useful, I don't think this should be weighed very highly in terms of contributions, as this would be considered standard practice in most similar problems.

- In short, the proposed contributions are minor in light of existing work, and more importantly, it is not clear from the results that they satisfy the goal stated in the introduction: to provide practical relevance in investigation of detected outliers.

---

> ### Author Response · Authors · 2023-03-19
> **Response to Reviewer vjgL (Part 1)**
>
> We thank the reviewer for their valuable comments on our paper and suggestions for improvements. We provide below a point-by-point answer to these comments, which we hope will help to clarify the contributions of our paper.
>
> > It is not clear from the paper how these "counterfactuals" are intended to be used and, arguably, their quality cannot be assessed without specifying this.
>
> Counterfactual explanation for time series anomalies provide insight on how features would need to be changed to be non-anomalous for the model. For instance, if a model detects anomalous values in medical records, counterfactual explanation may be used to assess what could be changed by the patient or physician to reduce the risk of encountering anomalous values in the future. We also note that this contrastive type of explanation  is often considered as better aligned with human needs than feature-based methods [1,2].
>
> > In the paper, explanations are given for three examples (Figure 1 & 2) but it is difficult to ascertain the practical value of the method from these examples since the anomalies and plausible counterfactuals are clear even without the model predictions.
>
> We agree that Figures 1 and 2, which only illustrate our explanations in the case of spike anomalies in univariate time series, do not provide a full insight on the value of our method. However, in Figures 3 and 5, we report examples of counterfactual explanations that may not be as intuitive as the previous examples. For instance, in the multivariate example in Figure 5, the spike at the sixth dimension is not smoothed in any of the counterfactual examples, therefore indicating that this spike is ``begnin" according to the anomaly detection model, contrary to the spikes at the seventh and eigth dimensions.
>
> > Moreover, there is no discussion about how to handle possibly erroneous predictions made by the model. Are explanations as useful when the anomaly detection model is wrong as when it is correct?
>
> In our work, we consider explanations for the correct predictions of a model (i.e., the True Positive anomalies), for which the model's user may use the counterfactuals to analyse or change the data. Nonetheless, counterfactual explanations can also be used to analyse erroneous predictions (i.e., False Positive anomalies), for debugging purposes or for improving the model. We have provided a preliminary experiment on applying our method to False Positive anoamlies in Appendix B2. We noted that for the latter, counterfactual examples are often easier to find and closer to the original time series. Therefore, in this context, counterfactual explanations may be used by the model's user to debug the model.
>
> > The ensemble is created by storing example inputs on the optimization path of gradient descent minimizing the same objective. This is unlikely to create meaningful diversity. For the type of large differentiable models that the authors target in this work, it is unlikely that the objective has a unique local (or global) minimum, and hence, there may be many counterfactuals that are fundamentally different but do not lay on the same path of gradient descent (since this will explore only one local minimum).
>
> We agree that our objective function may have multiple local minima for large models and high-dimensional time series, and one possibility for finding diverse examples would be to target these different local mimima. However, we found the latter approach difficult to achieve in  the early stages of our experimental study. In particular, for univariate time series, we observed that a random initialisation of the counterfactual example always led to the same solution of the optimisation problem. For multivariate time series, because of the difficulty of the optimisation problem, counterfactual examples could not be found unless the algorithm was initialised at the original anomalous windows or close to it.
>
> In contrast, we found that our strategy, which only explores one optimisation path, can provide counterfactual examples achieving different trade-offs between desirable properties and we provide an interpretation for this empirical fact. In our method, the hyperparameters weighting criteria in the  objective functions (1) and (2) are fixed for a data set, however, they may not provide the adequate trade-off for a particular instance of anomaly. Instead, along an optimisation path, we may find examples that would be the solutions of problems with different hyperparameters that may better suited for this particular anomaly. With a careful tuning of the learning rate of the SGD, we have found this strategy to provide a reasonable amount of diversity.
>
> References
>
> [1] T. Miller. Explanation in artificial intelligence: Insights from the social sciences, 2017. [arxiv](https://arxiv.org/abs/1706.07269)
>
> [2] E. Kumar et al.. Problems with shapley-value-based explanations as feature importance measures, 2020. [arxiv](https://arxiv.org/abs/2002.11097)

---

> ### Author Response · Authors · 2023-03-19
> **Response to Reviewer vjgL (Part 2)**
>
> > In 4.1.3, alternatives to this strategy are discussed, such as including a proximity penalty between examples or selecting optima for different random initializations. It is not clear from the paper in which scenario the proposed heuristic is preferable.
>
> We have not tested the alternative approach of solving an optimisation problem over multiple examples, but we expect the latter to be much more expensive to compute, and to lessen the relative importance of the other criteria with respect to the diversity property. The advantage of our approach is to achieve diversity without additional cost, which may be preferable for very large prediction models, for which an a complex optimisation objective could be unfeasible (see for instance [3]), or in contexts where diversity would not be the main desirable property. In other words, our method may be preferable in cases where it is enough for the end user to be shown different while relatively close values.
>
> > No explanations for high-dimensional are illustrated in the paper, nor were such explanations evaluated with humans interpreting them.
>
> We thank the reviewer for their comment. In our updated manuscript, we have moved illustrations of our method on the multivariate data sets (Figure 5) to the core of the paper.
>
> > Although generally useful, I don't think *[ the proposed visualization scheme]* should be weighed very highly in terms of contributions.
>
> We agree that the visualisation part is not the main contribution of our paper, although we had not seen it in previous works on diverse explanations.
>
> > In short, the proposed contributions are minor in light of existing work, and more importantly, it is not clear from the results that they satisfy the goal stated in the introduction: to provide practical relevance in investigation of detected outliers.
>
> It is true that our work combines ideas from prior work on time series perturbation and diverse counterfactual explanations. However, we believe that the combined solution -- diverse counterfactual explanations for time series anomaly detection models -- is novel and can be more useful for ML model users than existing explainability methods for this task. In particular, a counterfactual explanation allows to draw actionable insight on the time series, a property that it is not currently achieved by existing feature-saliency methods.
>
> > In the abstract, the authors claim to "show that our algorithm can produce valuable explanations". Valuable to whom? Valuable in what way?
>
> In Section 5.2 we described a complete set of metrics to capture the utility of our counterfactual explanations. Beyond these evaluation metrics, we would like to point out that we specifically use counterfactual explanations since these are better than alternatives like feature attribution methods like LIME and SHAP. A number of recent studies such as [4,5,6] have argued that counterfactual explanations better align with social and legal desiderata.
>
> > The chosen method *[for achieving diversity]* Is not well justified mathematically (there may be multiple local optima) or conceptually (it is not clear why humans would prefer this proposed kind of diversity).
>
> We agree that our method to provide diverse examples lacks some theoretical justification, but we found to empirically have good  properties, while, as previously mentioned, the random initialisation strategy did not work in practice.
>
> > Only one baseline is included in the experiments, and it is neither of the methods described in 4.1.3. Is there a reason these could not be used in the empirical evaluation?
>
> To the best of our knowledge, there is not yet a method for diverse counterfactual explanation in time series. The methods cited in Section 4.1.3 are designed for tabular or image data, therefore, we have deemed that including them in our experiments would require to adapt them to the time series context and would not be considered as existing baselines.
>
> > There is a large literature on adversarial examples, some of which go to great lengths to make adversarial examples plausible.
>
> We agree with the reviewer's point. In fact, some recent work [6] on connecting adversarial perturbation and counterfactual explanation shows that perturbations with perceptual loss (a form of plausibility constraint) resemble counterfactual explanations. We will update the text accordingly.
>
> References
>
> [3] D. Ley et al. Diverse, global and amortised counterfactual explanations for uncertainty estimates. [link](https://ojs.aaai.org/index.php/AAAI/article/view/20702)
>
> [4] T. Miller et al. Explainable ai: Beware of inmates running the asylum or: How i learnt to stop worrying and love the social and behavioural sciences, 2017. [arxiv](https://arxiv.org/abs/1712.00547)
>
> [5] S. Wachter et al. Counterfactual explanations without opening the black box: Automated decisions and the gdpr.
>
> [6] A. Elliott et al.. Explaining classifiers using adversarial perturbations on the perceptual ball.

---

### Review · Reviewer_oPeC · 2022-11-28

**Summary Of Contributions:**

**Contributions:**

* presenting several post-hoc `counterfactual ensemble explanation algorithms` that explain why ML predict certain time segments as anomalies in either univariate or multivariate time series
* adapting several metrics used for the evaluation of ensemble explanations in other data domains for time series data
* conducting a bunch of experiments in real-world datasets

**Audience:**

Yes

**Broader Impact Concerns:**

I do not have concerns about this part.

**Claims And Evidence:**

Yes

**Requested Changes:**

**Major concerns that have to be addressed:**

1. The algorithms do not consider contextual parts of given time windows, which make them quite limited and the authors should provide a better justification for this setting.

2. Several technical details are missing or not well justified

   a. Section 3.4 and Section 4.1.3, what is the final solution that is employed to ensure diversity? My understanding is just a sample from the candidate set without optimizing any metrics. What is the specific sample algorithm that is used?

   b. Why not use DTW in equation 1 to measure the closeness

   c. Equation 1, the authors should elaborate on the meaning of the plausibility through temporal smoothness

   d. Section 4.1.2, is the map matrix provided by end users? If my understanding is correct, I think this is not practical.

   e. Visualization is not well explained and is hard to follow. Please justify the design rationale and explain the visual encoding with concrete examples.

3. Evaluation part needs to be further improved

   a. it is necessary to conduct a user study to justify the necessity of showing counterfactual ensemble explanations and how they perceive the usefulness of explanation diversity.

   b. Implausibility 2 does not make sense when the original time series contains a lot of spikes. SMAP seems having many signals like this.

   c. Negative log-likelihood needs a further explanation.

   d. P5: "Note that we implicitly suppose that anomalies are not too close to each other so that the additional context window does not contain any anomaly" This is really depending on the datasets and the AD algorithms. Please discuss it with specific numbers in the experiment section and describe how the size of context window is picked, what are the impacts?

   e. In qualitative analysis (figure 2), I suggest the authors to replace one of the examples using a time series containing contextual anomalies whose values are not necessarily very high/low but may violate the trend patterns.

4. The usefulness of the insights from the experiments are limited. Though several explanation algorithms are presented to address different challenges in different scenarios, there are no explicit guidance of how to use them properly

**Other minor concerns:**

1. P2: "(a) knowing what can be changed in the input
   data to avoid encountering the anomaly again in the future". I suggest the authors to give some specific references to support this assumption. From my personal experience, domain experts always have a lot of concerns changing values in time series.
2. P4: "X_i denotes respectively the i-th row or the i-th **observation**." Does the "observation" here refer to "channel"?
3. P4: "we denote X - R(T x D) a time series with T time steps and D dimensions." This is very confusing. In the previous paragraph, X belongs to R(m x n). It seems the meanings of row and column are inverted.
4. P8: Sometimes PI_G is used and sometimes PI is used, which is confusing.
5. P13: please elaborate on "around 90 min".
6. Table 1, adding the average number of time stamps or anomalies per time series will be helpful.

**Strengths And Weaknesses:**

**Strengths:**

+ The paper is well-written and easy to follow
+ The idea of introducing contextual ensemble explanations in time series is good
+ The authors consider different scenarios and the proposed solutions are reasonable

**Weaknesses:**

+ The algorithms do not consider contextual parts of given time windows, which make them quite limited and the authors should provide a better justification for this setting.
+ Several technical details are missing or not well justified
+ Evaluation part needs to be further improved
+ The usefulness of the insights from the experiments are limited. Though several explanation algorithms are presented to address different challenges in different scenarios, there are no explicit guidance of how to use them properly

---

> ### Author Response · Authors · 2023-03-19
> **Response to Reviewer oPeC (Points 1 and 2)**
>
> We thank the reviewer for their valuable comments on our paper.
>
> > The algorithms do not consider contextual parts of given time windows, which make them quite limited and the authors should provide a better justification for this setting.
>
> We are not sure what the reviewer refers to with *contextual parts*. Is it a sub-window of the time series before an anomaly? We note that some anomaly detection models, such as NCAD, use a *context* window but this is not the case for all anomaly detection models, e.g., USAD. Our counterfactual method uses the context window in the former case to evaluate the anomaly score, however, a counterfactual example output by our method does not modify the contextual window, since the latter contains by definition the reference distribution and not the anomaly, i.e., does not need to be modified to be non-anomalous.
>
> > Several technical details are missing or not well justified.
>
> We would be happy to improve our paper by including the answers to the following comments in our updated manuscript.
>
> > a. Section 3.4 and Section 4.1.3, what is the final solution that is employed to ensure diversity? My understanding is just a sample from the candidate set without optimizing any metrics. What is the specific sample algorithm that is used?
>
> Our method does not have an explicit mechanism to ensure diversity in the set of counterfactual examples. However, we argue that the set of non-anomalous samples along the optimisation path can contain solutions of slightly different optimisation problems, achieving differientated trade-offs between multiple desirable properties of counterfactual examples. We therefore achieve diversity at a low cost, selecting a few number of counterfactuals along one optimisation path by regularly subsampling them by their iteration index.
>
> > b. Why not use DTW in equation 1 to measure the closeness?
>
> Computing the DTW distance is in general of complexity $O(L^2)$ for a time series of length $L$. It is therefore more costly to evaluate and to optimise than the Euclidean distance, which is of complexity $O(L)$, hence our choice of the latter in the objective (1) for practical considerations.
>
> > c. Equation 1, the authors should elaborate on the meaning of the plausibility through temporal smoothness.
>
> We first note that plausibility is in fact not easy to assess in our gradient-based method, since the latter does not rely on a generative model like in [1] -- which can be computationally expensive. Therefore, we use temporal smoothness as a convenient proxy measure, relying on the assumption that anomalies are in general less smooth than normal time series. In other words, a desirable property for a counterfactual example in our method is to be smoother than the original anomalous time series.
>
> > d. Section 4.1.2, is the map matrix provided by end users?
>
> No, the perturbation map $M$ in the DPE variant of our method is not provided by the user but by our method after optimising the objective (2). This map is the input of a fixed perturbation mechanism that outputs a counterfactual example. In other words, optimising the map matrix $M$ is equivalent to optimising with respect to the counterfactual example.
>
> > e. Visualization is not well explained and is hard to follow. Please justify the design rationale and explain the visual encoding with concrete examples.
>
> The visualization of our counterfactual set relies on the facts that every counterfactual example  has the same context window and a different anomaly score. In our figures, e.g., Figure 2, we plot the original time series (in blue) and the set of counterfactuals (e.g., in orange for the ICE variant), with a color scale indicating the anomaly score of each example (here between 0 and 1).
>
> References
>
> [1] Jana Lang, Martin Giese, Winfried Ilg, and Sebastian Otte. Generating sparse counterfactual explanations for multivariate time series, 2022.

---

> ### Author Response · Authors · 2023-03-19
> **Response to Reviewer oPeC (Points 3 and 4)**
>
> > Evaluation part needs to be further improved
>
> We agree that the empirical evaluation of our method could be improved in future work, in particular, regarding the practical evaluation of counterfactual ensemble explanations.
>
> > a. it is necessary to conduct a user study to justify the necessity of showing counterfactual ensemble explanations and how they perceive the usefulness of explanation diversity.
>
> A user study could indeed be relevant to assess the utility of the ensemble explanation. However, since such study cannot be easily conducted, we refer to previous work for highlighting the potential gains of diverse counterfactual explanations: for instance, [1] claim that  a diverse set of perturbed features provides different perspectives on how to answer a counterfactual query, an argument similarly stated in [2] and [3].
>
> b. Implausibility 2 does not make sense when the original time series contains a lot of spikes.
>
> We agree that Implausibility 2 (i.e., temporal smoothness) is not a useful metric for very unsmooth time series, all the more when anomalies are smoother than normal instances. However, there are also several contexts where anomalies correspond to (group of) spikes, while normal time series are quite smooth. We note temporal smoothness is only a proxy measure for plausibility, which is not easy to assess without a probabilistic generative model, and it may be inadequate in specific time series contexts.
>
> c. Negative log-likelihood needs a further explanation.
>
> Our gradient-free approach Forecasting Set fits a probabilistic forecasting model and generates samples from the forecast distribution, which likelihood can be evaluated. We also use its likelihood function to evaluate the plausibility of samples generated by our gradient-based approaches, DPE and ICE. In other words, the negative log-likelihood criterion corresponds to the negative log probability of the counterfactual examples, under the sampling model of Forecasting Set.
>
> d. P5: "Note that we implicitly suppose that anomalies are not too close to each other so that the additional context window does not contain any anomaly" This is really depending on the datasets and the AD algorithms. Please discuss it with specific numbers in the experiment section and describe how the size of context window is picked, what are the impacts?
>
> We first note that the size of the context window in the counterfactual example does not have any impact in our optimisation procedure since only the suspect window, where the anomaly is detected, is modified to construct counterfactual examples in our method. Nonetheless, the context window appears in our objective functions only because the anomaly detection model  may use it to compute the anomaly score, e.g., the NCAD model. Therefore, the size of the context window is a hyperparameter of the anomaly detection model, not our counterfactual explanation algorithm. Nonetheless, when the detection model only uses the "suspect" window to predict its anomaly score, we add a context window to the counterfactual suspect window *only for visualization purposes*. In this case, its size can be chosen in an *ad hoc* manner, depending on the frequency of anomalies in the data set.
>
> > e. In qualitative analysis (figure 2), I suggest the authors to replace one of the examples using a time series containing contextual anomalies whose values are not necessarily very high/low but may violate the trend patterns.
>
> We thank the reviewer for this suggestion and we would like to update our manuscript with an example of anomaly with non-extreme values, e.g., a frequency change in the seasonality of a time series.
>
> > The usefulness of the insights from the experiments are limited. Though several explanation algorithms are presented to address different challenges in different scenarios, there are no explicit guidance of how to use them properly
>
> In our experimental section, we have aimed at comparing\
> (i) gradient-based (DPE/ICE) and gradient-free (i.e., generative) (Forecasting Set) approaches for constructing counterfactual ensemble explanations;\
> (ii) different perturbation mechanisms (DPE and ICE);\
> (iii) sparse and non-sparse methods for multivariate time series anomalies.\
> We indeed do not provide a prescriptive approach on how to choose amongst the variants in practice, since the best choice may be problem-dependent. Nonetheless, we would like to update our manuscript with a qualitative description of the contexts in which we think that one variant may be preferred, e.g., when some domain knowledge about the regularity of the time series is known.
>
> **References**
>
> [1] B. Smyth and M. T. Keane. A few good counterfactuals: Generating interpretable, plausible and diverse counterfactual explanations.
>
> [2] P. Rodriguez et al. Beyond trivial counterfactual explanations with diverse valuable explanations.
>
> [3] A. S. Mothilal et al. Explaining machine learning classifiers through diverse counterfactual explanations.

---

### Review · Reviewer_tKZF · 2023-03-14

**Summary Of Contributions:**

In this paper, an explainability method for time series anomaly detection, namely Counterfactual Ensemble Explanation, is proposed. The proposed approach is model-agnostic and can be applied to any different detection model. Moreover, a sparse variant of the proposed method for high-dimensional time series anomalies is designed.

**Audience:**

Yes

**Broader Impact Concerns:**

N.A.

**Claims And Evidence:**

Yes

**Requested Changes:**

I would appreciate it if the authors can address the aforementioned issues.

**Strengths And Weaknesses:**

Pros:
1. This paper studies an interesting problem, i.e., explanations in the form of a set of diverse counterfactual examples are provided for time series anomaly detection.
2. The proposed method seems to be promising, and the authors have exhibited extensive experiment results.

However, I still have several concerns as follows.

Cons:
1. The authors should clearly clarify the challenges this work aims to solve in the Introduction Section. And the contributions in this work seem to be ambiguous, I suggest the authors clarify their contributions by using more logical words.

2. In Related Work, I suggest the authors provide more content to introduce anomaly detection in univariate time series and multivariate time series. There are some important time series anomaly detection works are missing, e.g., [1][2][3][4].

[1] Time series anomaly detection using temporal hierarchical one-class network.
[2] TimeAutoAD: Autonomous Anomaly Detection With Self-Supervised Contrastive Loss for Multivariate Time Series.
[3] Tranad: Deep transformer networks for anomaly detection in multivariate time series data.
[4] Clustering-based anomaly detection in multivariate time series data.

3. In Section 4.1.3, the analysis of optimization and complexity is informal. For example, what is your basis for selecting T? If you want to use SGD to guarantee convergence when the objectives are nonconvex, the T should be O(1/$\epsilon^2$). I believe that the authors should conduct more specific discussions about the setting of T. Moreover, according to your description, the algorithm seems to be a nested loop. I suggest the author conduct a standard complexity about the proposed algorithm, e.g., arithmetic complexity.

4. The authors should carefully modify the tables and figures. For example, Table 3 is suggested to move above the Section Discussion & Conclusion.

---

> ### Author Response · Authors · 2023-03-19
> **Response to Reviewer tKZF**
>
> We thank the reviewer for their valuable feedback and suggestions.
>
> > The authors should clearly clarify the challenges this work aims to solve in the Introduction Section. And the contributions in this work seem to be ambiguous, I suggest the authors clarify their contributions by using more logical words.
>
> We thank the reviewer for their comment and we would like to clarify the aims and contributions of our work in the following way.
>
> The **problem** that our paper aims to tackle is explaining the predictions of time series anomaly detection models.
>
> In our opinion, the **main challenges** are \
> (i) deep-learning models only provide an anomaly score for each time stamp, but their internal mechanism like the generative model or context windows, are neither user-facing nor easy to interpret;\
> (ii) for the task of anomaly detection, feature-based explanation methods like LIME and SHAP are ill-suited since they mark the model output -- that is, the anomalous timesteps -- as important features.
>
> In this context, **our contributions** are:\
> (i) the design of an optimisation objective for finding counterfactual examples for time series anomaly detection, a type of explanation that does not suffer from the drawbacks of feature-based explanations;\
> (ii) a method for obtaining diverse counterfactuals, that in particular allows to understand the model's behavior across a broad range of values in a neighborhood of the original input;
> (iii) a sparse variant for multivariate time series that enables the users to visualise low-dimensional explanations.
>
> > In Related Work, I suggest the authors provide more content to introduce anomaly detection in univariate time series and multivariate time series. There are some important time series anomaly detection works are missing.
>
> We thank the reviewer for this suggestion and we propose to update our manuscript  by adding the following paragraph at the beginning of  our related work section, to introduce the task of anomaly detection in time series.
>
> "Time series anomalies can represent a risk, a failure, or an abnormal state of a system, that requires intervention. Therefore, highly-performing deep-learning models have been developed to detect them, for instance using deep support vector data description [1] or transformer networks [2]. However, most of these models provide little interpretation or diagnosis of detected anomalies. In particular, finding the root cause of an anomaly often consists in identifying channels of the time series containing anomalous features [3,4]. However, a user of an anomaly detection model may be interested in understanding the time series *normal* distribution, and how the anomaly could be avoided in the future."
>
> > In Section 4.1.3, the analysis of optimization and complexity is informal. For example, what is your basis for selecting T? If you want to use SGD to guarantee convergence when the objectives are nonconvex, the T should be $\mathcal{O}(1/\epsilon^2)$. I believe that the authors should conduct more specific discussions about the setting of T.
>
> We thank the reviewer for their comment. We first note that our goal is not necessarily to reach a local minima of the objective, but essentially to find (diverse) examples which prediction scores are below the anomaly detection threshold along the optimisation path.
>
> In our experiments, we have run the SGD for $T=1000$ iterations, which we empirically found is enough for the algorithm to stabilise around a local minimum and essentially to explore one optimisation path. However, an improvement of our method could be to use an early stopping strategy once a "good enough" local minimum has been reached.
>
> > Moreover, according to your description, the algorithm seems to be a nested loop. I suggest the author conduct a standard complexity about the proposed algorithm, e.g., arithmetic complexity.
>
> We are not sure what the reviewer means by 'nested loop'. Our algorithm consists of only one run of SGD for $T$ iterations.
>
> > The authors should carefully modify the tables and figures.
>
> In our updated manuscript, we have modified our tables and figures to appear in the section they are related to.
>
> References
>
> [1] Lifeng Shen, Zhuocong Li, and James Kwok. Timeseries anomaly detection using temporal hierarchical one-class network. I
>
> [2] Shreshth Tuli, Giuliano Casale, and Nicholas R. Jennings. Tranad: Deep transformer networks for anomaly detection in multivariate time series data.
>
> [3] Chuxu Zhang, Dongjin Song, Yuncong Chen, Xinyang Feng, Cristian Lumezanu, Wei Cheng, Jingchao Ni, Bo Zong, Haifeng Chen, and Nitesh V Chawla. A deep neural network for unsupervised anomaly detection and diagnosis in multivariate time series data.
>
> [4] Ailin Deng and Bryan Hooi. Graph neural network-based anomaly detection in multivariate time series.

---

### Decision · Action_Editors · 2023-04-22

**Recommendation:** Reject

**Comment:**

After the review and rebuttal process, all 3 reviewers recommended rejection of this paper. A major concern is that the authors did not update their manuscript during the rebuttal process which hindered the reviewers' capabilities to evaluate the proposed revisions properly. Besides, there are still major issues in terms of evaluation of the framework. Reviewers would like more quantitative evaluations, e.g. a user study, to demonstrate the usefulness of the proposed framework. Besides, it would be nice to have a case study application of the proposed framework. If all these tasks would have been done, then the authors would be encouraged to make a major revision and resubmit this work for a new round of reviews.

**Audience:**

Yes

**Claims And Evidence:**

This paper proposes an approach to generate counterfactual explanations for anomaly detection in time series. The evaluation of the paper is weak as it mostly only contained qualitative examples, without enough quantitative results to back it up.